# *Cobrançosa* Table Olive Fermentation as per the Portuguese Traditional Method, Using Potentially Probiotic *Lactiplantibacillus pentosus* i106 upon Alternative Inoculation Strategies

**Joana Coimbra-Gomes** [1,2], **Patrícia J. M. Reis** [1,2,*], **Tânia G. Tavares** [1,2], **Afonso A. Silva** [1,2], **Eulália Mendes** [3], **Susana Casal** [3], **Francisco Xavier Malcata** [1,2] and **Angela C. Macedo** [1,2,4]

1    LEPABE—Laboratory for Process Engineering, Environment, Biotechnology and Energy, Faculty of Engineering, University of Porto, Rua Doutor Roberto Frias, 4200-465 Porto, Portugal
2    ALiCE—Associate Laboratory in Chemical Engineering, Faculty of Engineering, University of Porto, Rua Doutor Roberto Frias, 4200-465 Porto, Portugal
3    LAQV/REQUIMTE, Laboratory of Bromatology and Hydrology, Faculty of Pharmacy, University of Porto, Rua de Jorge Viterbo Ferreira 228, 4050-313 Porto, Portugal
4    UNICES—Research Unit in Management Sciences and Sustainability, University of Maia, Avenida Carlos de Oliveira Campos, 4475-690 Maia, Portugal
*    Correspondence: pmreis@fe.up.pt

**Abstract:** Spontaneous fermentation of table olives, as per a traditional Mediterranean process, is still performed empirically; hence, final product quality is somewhat unpredictable. Our main goal was to validate an endogenous (potentially probiotic) lactic acid bacterium strain in *Cobrançosa* table olives as a vector for a more standardized process, further adding commercial value to the olives themselves. The traditional Portuguese fermentation process typically consists of two stages: sweetening, when olives are periodically washed with spring water to different proportions, and salting, when water is no longer changed, but salt is gradually added to the brine, up to 7–10% (*w/w*). *Lactiplantibacillus pentosus* i106 was inoculated as follows: (plan A) 2020/21 harvest, with 0, 3, 5, and 7% (*w/v*) NaCl, without sweetening; (plan B) 2020/21 harvest, with 5 and 7% (*w/v*) NaCl, during salting and sweetening; and (plan C) 2019/20 harvest, with 5% (*w/v*) salt, and sweetening and salting. Microbiological, physical, and biochemical evolutions were monitored for 8 months, and final nutritional and sensory features were duly assessed. Compared to the control, lactic acid bacteria (LAB) predominated over yeasts only if deliberately inoculated; however strain viability was hindered above 5% (*w/w*) NaCl, and LAB inhibited enterobacteria. Degradation of (bitter) oleuropein to hydroxytyrosol and verbascoside was faster upon inoculation. Color-changing olives from the 2020/21 harvest exhibited higher fat content and lower water content compared to green ones (2019/20 harvest), and different salt levels and inoculation moments produced distinct sensory properties. The best protocol was plan C, in terms of overall eating quality; hence, the addition of *Lpb. pentosus* i106 provides benefits as a supplementary additive (or adjunct culture), rather than a starter culture.

**Keywords:** table olives; lactic acid bacteria; probiotics; fermentation; starter culture; microbiological profiles; physicochemical profiles; nutritional analysis; sensory analysis

## 1. Introduction

Traditional fermented foods are an important part of human cultural heritage and diet and have been widely consumed since ancient times, as well as being used as a form of preservation, increasing the safety of existing food resources. Table olives are among the most well-known and widely consumed fermented vegetable products in Mediterranean countries [1].

Unripe olives are hardly edible due to their high content of phenolic compounds, especially oleuropein, responsible for drupe bitterness [1]. Hence, table olives must undergo a series of preliminary physicochemical and microbiological transformations prior to consumption, such as fermentation in brine [2]. Several types of table olives have been singled out according to the degree of ripeness, and their mode of preparation varies from country to country [2,3]. Upon harvest, table olives may be readily submerged in brine, typically solutions of 6–10% (*w/v*) NaCl [4], or may instead undergo alkaline treatment with NaOH, controlled salt percentage, and low pH. Darkened table olives are treated directly or preserved in brine, oxidized, washed, and packed in cans, hermetically sealed and sterilized to prevent any fermentation, hence guaranteeing product safety, since the growth of pathogens and spoilage microorganisms is avoided [5]. After having undergone either of the above treatments, table olives are preserved in brine as per their specific characteristics, in dry salt under modified atmosphere, or even in a solution of preservative or acidifying agent(s); they may also be subjected to pasteurization to extend their shelf life [5]. Such fermentation parameters as pH or temperature are germane, yet the presence of salt in the medium is an important factor for food preservation in that it affects fermentation rate; at the same time, the latter contributes to taste and texture improvement, even though a high salt level will impair the healthy image built around this fermented food [6,7].

*Cobrançosa* cultivar, native to Portugal, has for ages been subjected to natural fermentation after harvest and before full maturation; hence, a variable color, ranging from green to color-changing, is normally found. Upon careful removal of rotten drupes, stalks, and leaves, the olives undergo a first stage of processing, termed the sweetening stage by the local producers, which holds bitterness removal as its major goal. During this period, table olives are washed periodically with spring water in different proportions and kept thereafter in water for 4–6 months. The period between consecutive renewals of water ranges from 1 week up to 2 months. The salting stage (so named also by the local producers) comes afterward; during this period, the water is no longer changed until the product is ready for the market, but salt is gradually added to the brine, up to 7–10% (*w/w*) by the time of sale [2,8]. A full characterization of this processing method and of typical microbiological and physicochemical profiles throughout time has been reported by Reis et al. [8]. The fermentation process starts spontaneously and is strongly influenced by the olive cultivar itself, its indigenous microbiota, and such methodological factors as fermentation temperature and salt concentration of brines [9]. This type of fermentation leads to somewhat unpredictable and variable quality of the final product, in addition to the susceptibility to the growth of undesirable microorganisms [10].

Diverse microbial populations are involved in olive fermentation, chiefly lactic acid bacteria (LAB) and yeasts; LAB are primarily responsible for brine acidification due to lactic acid production, which leads to a decrease in the pH and an increase in the microbiological stability of the final product, thus supporting an extended shelf life [1].

Over the years, it has become important to retrieve starter cultures from the native microbiota of olive fermentations and then study their technological characteristics and probiotic potential. If ingested in appropriate quantities, probiotics can exert several beneficial effects upon human health; this has led the food industry to consider them, further driven by a growing demand by the market [2]. Appropriate microbial starters should exhibit a few specific biotechnological and safety traits, namely: easy and rapid adaptation to the brine environment (e.g., temperature, pH, and phenolic profile); rapid dominance over indigenous microbiota; rapid brine pH reduction (e.g., via synthesis of organic acids); strong enzymatic activity, suitable for enhancing the sensory features of the final product; and rapid degradation of oleuropein (e.g., via adventitious β-glucosidase and esterase), thus making the product eventually edible [11].

In the latest decade, several studies have focused on the addition of starter cultures with multifunctional potential to various types of table olives; this includes those treated via the Spanish style [3,12] or even by natural fermentation [13,14], but not with Portuguese cultivars, classically known for their unique bouquet. Since production of *Cobrançosa* table

olives starts with a sweetening stage (thus making it different from other processes abroad), testing of alternative strategies for inoculation appears logical, especially at distinct stages throughout the fermentation process (sweetening, salting, and ready-to-eat).

Therefore, the main goal of this study was to assess and validate, at bench scale, the use and viability of *Lpb. pentosus* i106 as a starter and/or potentially probiotic culture during fermentation of *Cobrançosa* table olives; hence, three inoculation approaches were considered (during sweetening, salting, and ready-to-eat times) in terms of their effects upon microbiological and physicochemical profiles of the final product. The aforementioned LAB strain, originally isolated from *Cobrançosa* table olives [8], was previously tested for its technological properties (such as acid and salt tolerances, survival at different temperatures, and degradation of oleuropein) and probiotic potential in vitro [15,16]. It is important to remember that the definition of a dedicated microbial starter culture will aid in reducing spoilage risk while allowing eventual optimization of processing and compression of the period taken by fermentation and achievement of a final product with higher quality, i.e., bearing putatively improved functional and sensory properties beyond basic nutritional features.

## 2. Materials and Methods

### 2.1. Bacterial Strains and Growth Conditions

*Lactiplantibacillus pentosus* strain i106, previously isolated from *Cobrançosa* table olives and brines [8] and selected for its potentially probiotic features [15,16], was used in this study. Prior to the experiments, and following the methodology reported by Blana et al. [17], this strain was revived from a stock culture (in 15% glycerol) at $-80$ °C, subcultured in 10 mL of de Man Rogosa Sharpe (MRS) broth (VWR Chemicals, Leuven, Belgium) with 0 or 3% ($w/v$) NaCl, and incubated at 37 °C for 8–10 h. Afterwards, an aliquot was transferred into 90 mL of fresh MRS broth with 0, 3, or 5%($w/v$) NaCl (depending on the trial) to obtain an initial Optical Density at 600 nm ($OD_{600nm}$) of ca. 0.1 (UV-1800 spectrophotometer, Shimadzu, Duisburg, Germany). After overnight incubation at 30 °C, a "working medium", prepared with mineral water, 2% peptone from meat (VWR), 2% glucose (Sigma-Aldrich, St. Louis, MO, USA), and 0, 3, 5, or 7% ($w/v$) NaCl (depending on the trial), was inoculated with an aliquot of the overnight culture to give an initial $OD_{600nm}$ of ca. 0.2 and incubated for 5–6 h at 37 °C. Finally, trial vessels were inoculated to an initial $OD_{600nm}$ of ca. 0.1. Initial LAB concentration should reach $10^8$ CFU/mL in each vessel. Gradual increases in NaCl concentration along the inoculum preparation are essential to allow adaptation of the strain to the saline environment prevailing in actual brine [18]. For microbiological monitoring of stock and working cultures, samples were serially diluted in 0.85% ($w/v$) sterile saline solution and spread plated in triplicate on MRS agar. Counts were produced by 48 h incubation at 30 °C by resorting to a colony counter (Scan® 100, VWR, Milano, Italy). Results were set as means of three determinations.

### 2.2. Table Olives Samples

Olives of cultivar *Cobrançosa* were provided by a well-referenced producer (Trás-os-Montes, northeast of Portugal) who resorts to the natural fermentation method; for the experiments, 5 L vessels of 4 kg of olives with water were supplied (plan A, ten vessels; plan B, six vessels) on the day following harvest, along with brine (5% salt) (plan C, four sealed vessels). While table olives for plan A and B were harvested in 2020/21, those for plan C came from an earlier harvest, 2019/20. All vessels were semi-covered and stored at room temperature (ca. 20 °C).

### 2.3. Experimental Design

According to Figure 1, table olive fermentation from designs A and B were monitored over 35 weeks (245 days), while those from design C were monitored over 10 weeks (65 days). In plan A, the addition of strain *Lpb. pentosus* i106 to 4 independent trials was considered: table olive water with no salt (A0), obtained exactly one week after harvest by the producer, and brines with 3 (A3), 5 (A5), and 7% ($w/w$) (A7) commercial kitchen salt. In

plan B, the addition of *Lpb. pentosus* i106 to the brine was tested at 5 (B5) and 7% (*w/w*) (B7) salt, during the salting process of fermentation, i.e., obtained after sweetening in loco by the producer. In plan C, the *Lpb. pentosus* i106 addition was assessed in ready-to-eat table olives (C5), i.e., obtained after sweetening and salting in loco by the producer. Although only one trial was inoculated at each inoculation time, salt was added to all vessels. It should be stressed that no water renewal had been originally planned in plan A; however, by the time of inoculation with 3% (*w/w*) salt, the water was found to be no longer transparent, most likely due to the diffusion of semi-soluble compounds from the olive pulp. As suggested by the producer, a new water renewal was carried out in trials A3, A5, and A7. The concomitant addition of salt and of the potentially probiotic strain i106 was planned so as to reach the best compromise between the results and conclusions reported in previous work [8] and to minimize brine agitation, according to traditional practices followed by the producers. Plan C lasted only one month because of the narrow availability to meet of the taste panel members and because addition of said probiotic after packaging would compromise its viability (should be used as a post-biotic, as further discussed in future research). For each plan, a control trial was also monitored (AC, BC, and CC), differing from experimental trials as per the addition of the potentially probiotic LAB strain. All trials were run in duplicate.

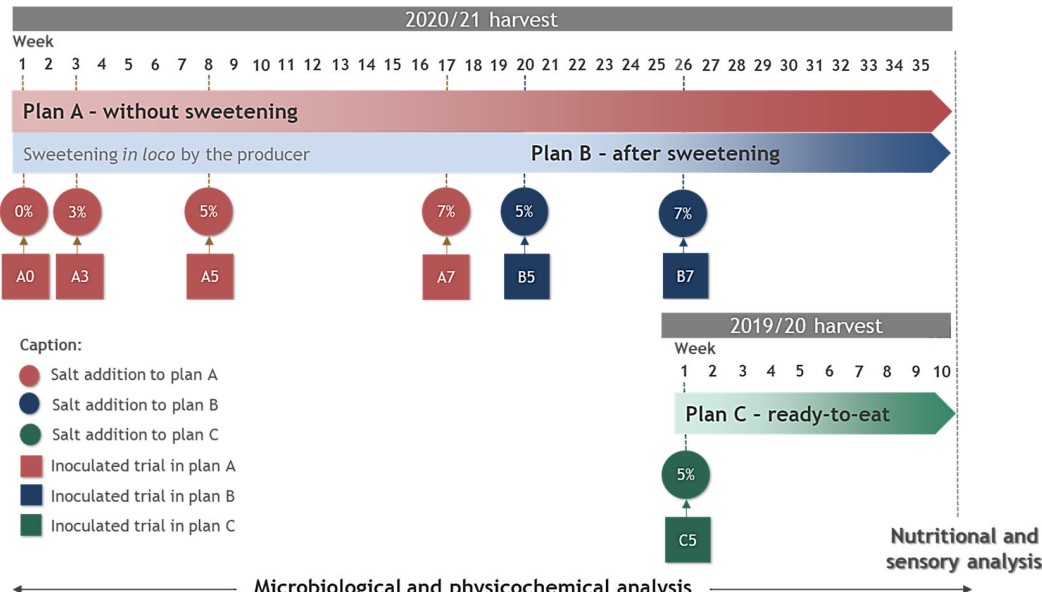

**Figure 1.** Scheme of experimental design followed, showing the inoculation times of *Lpb. pentosus* i106 under the different strategies (plans A, B, and C) and salt addition to vessels with *Cobrançosa* olives.

### *2.4. Microbiological Analysis*

Under an aseptic environment, ca. 20 table olives and sufficient brine to cover them were randomly collected from each vessel using previously sterilized utensils. Twelve pitted olives (around 40 g) were placed in a sterile bag with a side filter, together with 100 mL of saline solution (0.85%), and rubbed in a Stomacher Star Blender LB 400 (VWR, Milano, Italy) for 60 s; the filtered liquid was then poured into a sterile Falcon flask. Both the filtered liquid and brine were directly used for further dilution and duly plated. The method of Reis et al. [8] was followed for the enumeration of yeasts and LAB in the samples, while the enumeration of Enterobacteriaceae by pour plating followed Abriouel et al. [19]. Lactic acid bacteria and yeast growth was monitored by spread plating brine and pulp samples from table olives in MRS agar with 0.4 g/L sodium azide (VWR) followed by incubation for 48 h at 30 °C and Rose Bengal with 0.1 g/L chloramphenicol (RBC) agar (VWR) followed by incubation for by 5 d at 25 °C, respectively. To monitor Enterobacteriaceae growth, the samples were pour plated in Violet Red Bile Glucose (VRBGA) agar (VWR) and incubated

for 24 h at 30 °C. The results were expressed as log CFU/mL for brine and as log CFU/g for olive samples.

### 2.5. Physicochemical Analysis

Physicochemical analyses of table olive brines included salinity (% *w/v*), pH, and titratable acidity (lactic acid % (*w/v*)), as well as phenolic compound concentrations (mg/L), namely oleuropein, verbascoside, hydroxytyrosol, and tyrosol; the techniques used were as described elsewhere [9].

### 2.6. Nutritional Analysis

Prior to nutritional analysis, table olive pits were removed, and the pulp was milled (8000 rpm; 20 s) in a Knife Mill Grindomix GM 200 (Retsch), thus generating a paste. All procedures requiring sample weighing resorted to an Analytical Balance (KERN AES). Nutritional composition of samples was determined according to AOAC Official Methods [20], including moisture (925.40), total extractable fat (948.22), crude protein (920.152), total ash (940.26), and total dietary fiber (985.29) after fat extraction. Carbohydrate content was estimated by difference. Sodium chloride content was determined via titration with $AgNO_3$ (Mohr method), according to Fernandez-Diez et al. [21]. The energy value, expressed in kcal/100 g of pulp, was calculated following the Atwater system, using factor 4 for protein and carbohydrates, 9 for extracted lipids, and 2 for dietary fiber. Fatty acids were evaluated by gas chromatography, according to European Commission Regulation (EEC 2568/91, of 11th July) and as described by Rodrigues et al. [22]. The composition was expressed as relative abundance in a percentage.

### 2.7. Sensory Analysis

Following Anagnostopoulos et al. [23], table olive samples from all plans were evaluated by the end of the process by 12 panel members of a Mirandela local SME, consisting of 5 males and 7 females, aged from 40 to 60 years old, according to Regulation COI/OT/MO No 1/Rev.3 [24]. The results were presented in radar charts.

### 2.8. Statistical Analysis

All graphics and calculations of means and standard deviations were produced in Excel (Microsoft® Office 2021). A one-way analysis of variance (ANOVA), followed by Tukey's post hoc means multiple comparisons, were performed for physicochemical and nutritional characteristics, as well as overall eating quality scores, at the 5% level of significance, using IBM SPSS 27.0 software (IBM, Armonk, NY, USA). Letters in mean comparisons were ascribed as proposed by Piepho [25].

## 3. Results

### 3.1. Microbiological Monitoring

Changes in viable numbers of LAB, yeasts, and Enterobacteriaceae in brine and table olives, during fermentation of *Cobrançosa* table olives under the three alternative plans, were investigated; see Figure 2 (Plan A), Figure 3 (Plan B), and Figure 4 (plan C).

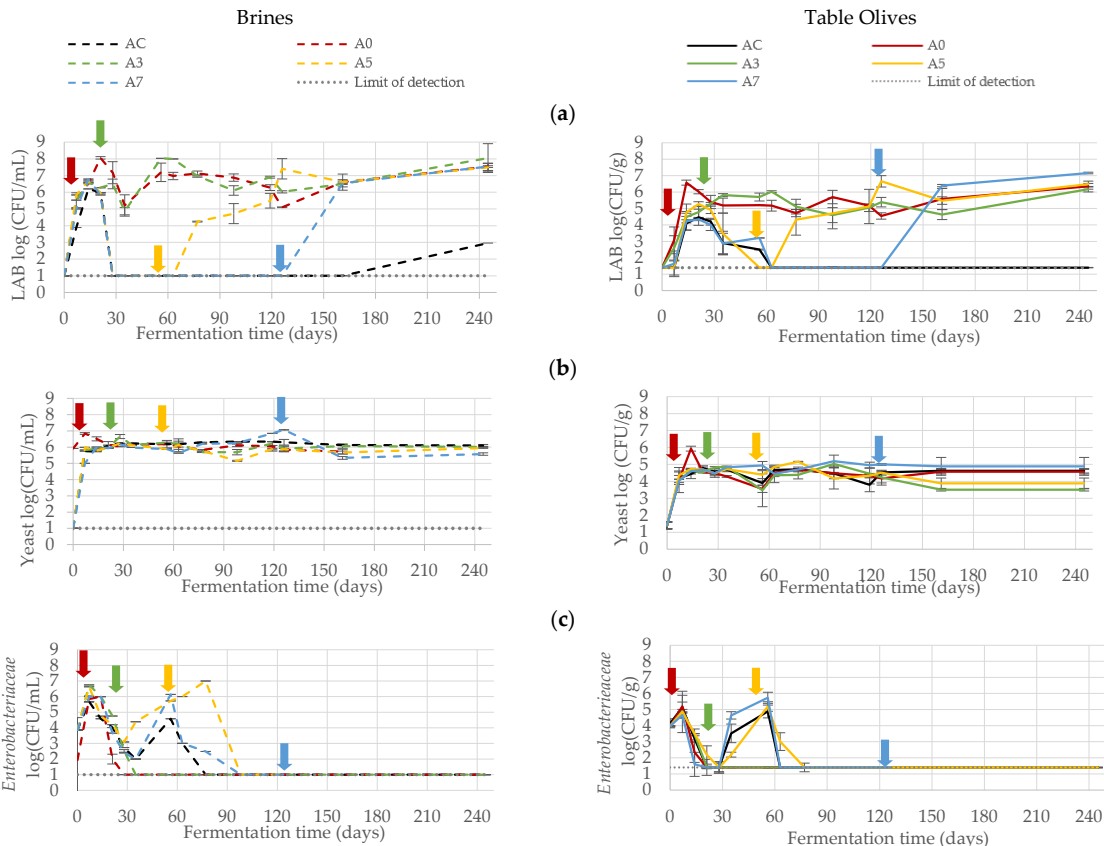

**Figure 2.** Evolution of viable numbers of (**a**) lactic acid bacteria, (**b**) yeasts, and (**c**) Enterobacteriaceae, in (**left**) brines and (**right**) table olives, under plan A. Arrows indicate the moment of inoculation with *Lpb. pentosus* i106 and the corresponding salt addition to all vessels.

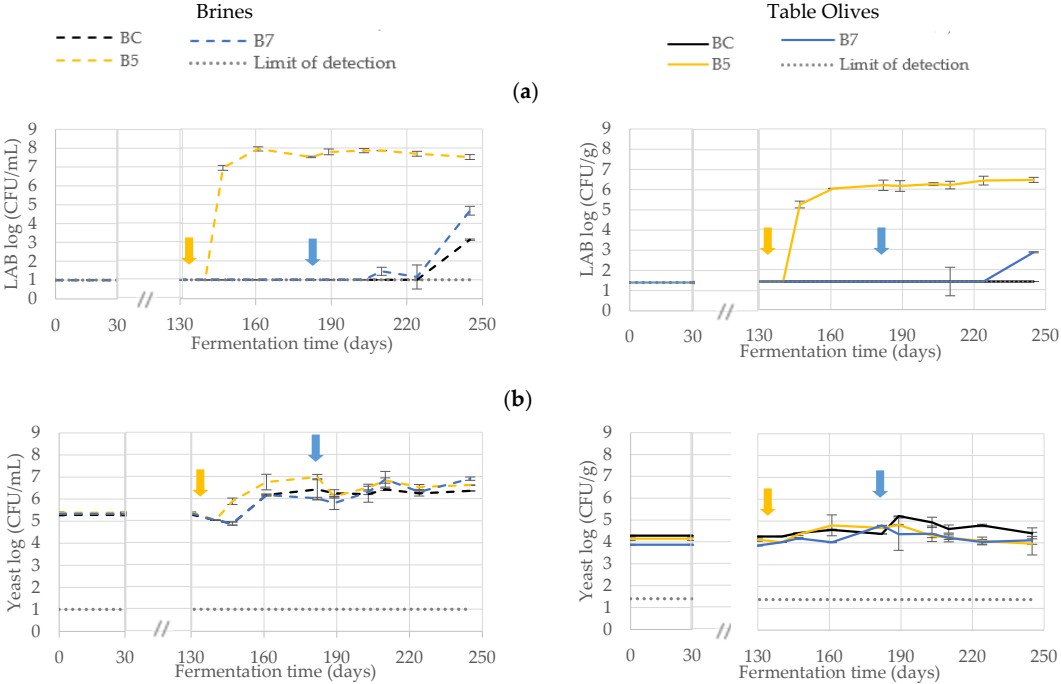

**Figure 3.** Evolution of viable numbers of (**a**) lactic acid bacteria and (**b**) yeasts, in (**left**) brines and (**right**) table olives, under plan B. Arrows indicate the moment of inoculation with *Lpb. pentosus* i106 and the corresponding salt addition to all vessels.

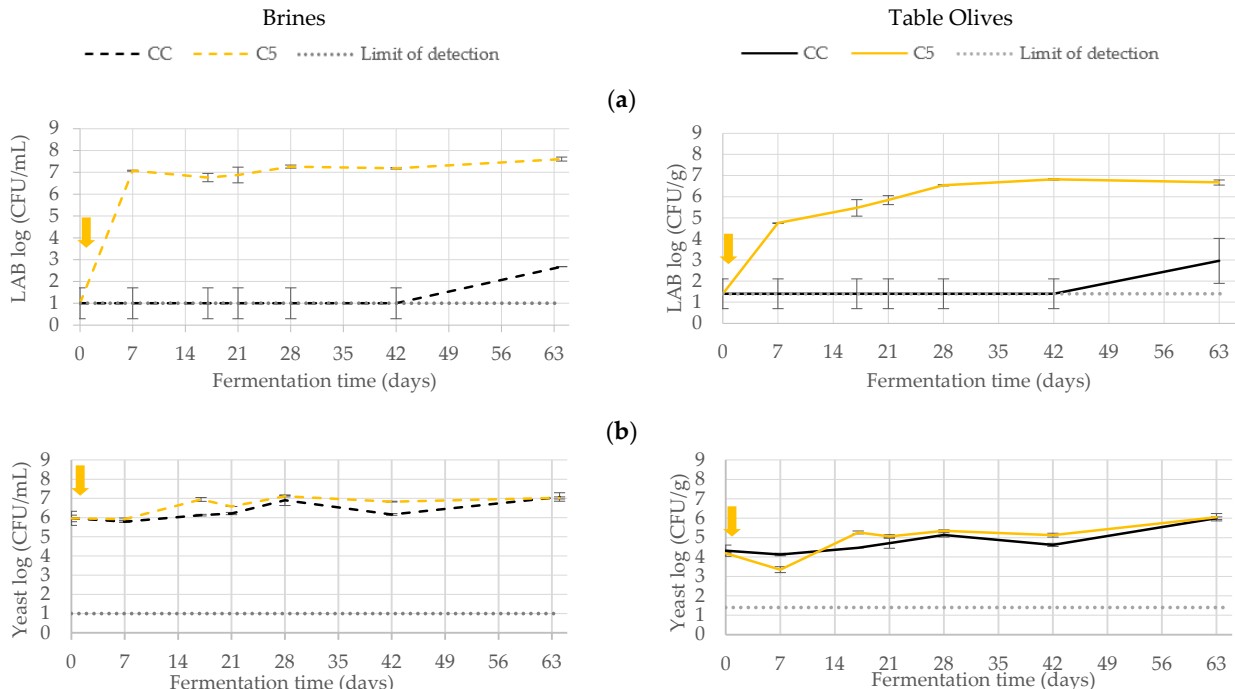

**Figure 4.** Evolution of viable numbers of (**a**) lactic acid bacteria and (**b**) yeasts, in (**left**) brines and (**right**) table olives, under plan C. Arrows indicate the moment of inoculation with *Lpb. pentosus* i106 and the corresponding salt addition to all vessels.

Under plan A (Figure 2), LAB population numbers (Figure 2a) remained high, at ca. 7 log CFU/mL in brine and ca. 6 log CFU/g in drupes, for the trials with early addition of the potentially probiotic strain (A0 and A3). In batches where strain i106 was added later (trials A5 and A7), the LAB population was initially high and then underwent a substantial decrease, before eventually recovering some time after the addition of the potentially probiotic strain to levels similar to those of the other batches. Regarding the control batches (AC), after some initial growth, LAB populations decreased and remained below the detection limit (1 log CFU/mL in brine and 1.4 log CFU/g in drupes); however, by the end of the experiment, some growth was observed in the brine. Regarding yeasts (Figure 2b), and despite an initial increase, their population eventually stabilized at 6 log CFU/mL and 5 log CFU/g in brine and drupes, respectively. As expected, the Enterobacteriaceae population (Figure 2c) decreased with time, until reaching values below the detection limit, both in brine and drupes.

Regarding plan B (Figure 3), while no Enterobacteriaceae counts were recorded, substantial differences were found between trials regarding the LAB populations (Figure 3a): AC presented LAB counts below the detection limit throughout the whole experiment, except by the end in the brine; some growth was observed, reaching ca. 3 log CFU/mL. A5 showed LAB growth to 7–8 log CFU/mL in brine and 6–7 log CFU/g in drupes 7 days after the addition of the potentially probiotic strain, followed by a plateau until the end of the experiment; finally, A7 exhibited a slower growth, with some LAB growth after 30 days, but only reached 5 log CFU/mL in brine and 3 log CFU/g in drupes. Regarding yeast populations (Figure 3b), no major differences were found between trials or during the experiment for the drupe (4–5 log CFU/g); in brine, a growth of 1–2 log CFU/mL was observed by 140 days, with numbers stabilizing at 6–7 log CFU/mL, also with no major differences between trials.

Similar to plan B, none of the trials under plan C (Figure 4) held Enterobacteriaceae cells, yet a substantial increase in LAB populations (Figure 4a) was observed along the process after strain i106 addition, both in brine (up to almost 8 log CFU/mL) and drupes (up to almost 7 log CFU/g). Regarding the control trial (CC), LAB populations started

to grow at 40 days of fermentation, reaching 2.5 log CFU/mL in brine and 3 log CFU/g in drupes. Concerning yeast populations (Figure 4b), no major difference was observed in brine among trials, nor throughout the experiment; on the other hand, a small growth (2 log CFU/g) was detected in drupes when comparing the end to the beginning of the experiment, but no differences were found among trials.

### 3.2. Physicochemical Monitoring

### 3.2.1. Salt Content, pH, and Total Titratable Acidity

Salt content, pH, and total titratable acidity in brine were ascertained under the different plans at some noteworthy times of fermentation, i.e., at the beginning of each experiment (time 0), one week after the addition of each salt level and *Lpb. pentosus* i106, and finally at the end of the experiment.

Under plan A (Table 1), salt content among trials did not differ significantly ($p < 0.05$) for each time of experiment, only differing within each trial throughout time as salt was added. Regarding pH values, all trials began at 5.2, but two patterns of evolution were detected afterwards: trials where *Lpb. pentosus* i106 was added with a low salt concentration (A0 and A3) exhibited a significant ($p < 0.05$) decrease (3.8 and 3.6, respectively) when compared to the control (4.4) at early stages of fermentation, then they approximated the control (ca. 4.1) around 3 months and maintained that level (ca. 4.7) until the end; in trials where *Lpb. pentosus* i106 was added with a medium/high salt concentration (A5 and A7), the pH profile was similar between the two trials and control (ca. 4.1) until ca. 3 months, but after the salt content reached 5%, pH tended to increase (up to 5.1) in trials A5 and A7, while the others stabilized (ca. 4.7, as mentioned before). At start, acidity was ca. 0.16% in all trials; then it is increased until reaching a maximum value depending on the trial, followed by a reduction down to ca. 0.20%. Trials A5 and A7 exhibited a decrease in TTA at the early stages of fermentation (of 0.10% by 28 days), when LAB could not be detected; this pattern was not found in the control, either for A0 or A3, where LAB were detected.

**Table 1.** Evolution in physicochemical features (salt content, pH, and total titratable acidity) of brine at selected times of fermentation under plan A.

| | Trial | Fermentation (Days) | | | | | | |
|---|---|---|---|---|---|---|---|---|
| | | 0 | 14 | 28 | 63 | 126 | 245 | *p*-Value |
| Salt (%) | AC | 0.00 ± 0.00 A | 0.00 ± 0.00 A | 2.90 ± 0.14 aB | 5.00 ± 0.00 C | 6.70 ± 0.28 D | 5.50 ± 0.00 aC | <0.001 |
| | A0 | 0.00 ± 0.00 A | 0.00 ± 0.00 A | 3.80 ± 0.28 bB | 5.00 ± 0.00 C | 6.75 ± 0.35 E | 5.80 ± 0.00 abD | <0.001 |
| | A3 | 0.00 ± 0.00 A | 0.00 ± 0.00 A | 3.00 ± 0.00 aB | 5.00 ± 0.00 C | 6.25 ± 0.35 E | 5.65 ± 0.07 abD | <0.001 |
| | A5 | 0.00 ± 0.00 A | 0.00 ± 0.00 A | 2.85 ± 0.07 aB | 5.00 ± 0.00 C | 6.00 ± 0.00 D | 5.95 ± 0.21 bD | <0.001 |
| | A7 | 0.00 ± 0.00 A | 0.00 ± 0.00 A | 3.00 ± 0.00 aB | 4.90 ± 0.14 C | 6.80 ± 0.28 E | 5.50 ± 0.00 aD | <0.001 |
| *p*-value | | 1.000 | 1.000 | 0.006 | 0.486 | 0.124 | 0.023 | |
| pH | AC | 5.20 ± 0.00 | 4.56 ± 0.01 | 4.40 ± 0.03 b | 4.44 ± 0.03 | 4.20 ± 0.07 ab | 4.56 ± 0.06 a | 0.082 |
| | A0 | 5.20 ± 0.00 E | 4.93 ± 0.01 D | 3.83 ± 0.01 aA | 3.95 ± 0.04 A | 4.22 ± 0.06 abB | 4.65 ± 0.13 abC | <0.001 |
| | A3 | 5.20 ± 0.00 D | 4.62 ± 0.01 C | 3.87 ± 0.11 aAB | 3.61 ± 0.03 A | 4.09 ± 0.09 aB | 4.83 ± 0.09 abcC | <0.001 |
| | A5 | 5.20 ± 0.00 C | 4.60 ± 0.05 B | 4.40 ± 0.03 bB | 4.54 ± 0.03 B | 4.08 ± 0.02 aA | 5.08 ± 0.16 bcC | <0.001 |
| | A7 | 5.20 ± 0.00 B | 4.60 ± 0.01 A | 4.42 ± 0.02 bA | 4.60 ± 0.01 B | 4.39 ± 0.08 bA | 5.19 ± 0.14 cB | <0.001 |
| *p*-value | | 1.000 | 0.584 | <0.001 | 0.532 | 0.030 | 0.010 | |
| TTA * | AC | 0.15 ± 0.04 A | 0.18 ± 0.01 bA | 0.10 ± 0.02 bA | 0.23 ± 0.03 aA | 0.40 ± 0.06 abB | 0.23 ± 0.01 A | 0.006 |
| | A0 | 0.16 ± 0.01 A | 0.40 ± 0.00 dB | 0.56 ± 0.01 cC | 0.54 ± 0.03 bBC | 0.46 ± 0.04 bBC | 0.27 ± 0.04 BC | <0.001 |
| | A3 | 0.17 ± 0.02 A | 0.18 ± 0.00 bA | 0.19 ± 0.01 bA | 0.59 ± 0.01 bD | 0.42 ± 0.04 abC | 0.19 ± 0.03 B | <0.001 |
| | A5 | 0.15 ± 0.01 AB | 0.16 ± 0.01 aAB | 0.11 ± 0.00 aA | 0.20 ± 0.02 aAB | 0.51 ± 0.01 bC | 0.17 ± 0.03 B | <0.001 |
| | A7 | 0.18 ± 0.01 AB | 0.20 ± 0.00 cB | 0.09 ± 0.00 aA | 0.18 ± 0.03 aAB | 0.27 ± 0.03 aB | 0.14 ± 0.01 B | 0.004 |
| *p*-value | | 0.665 | <0.001 | <0.001 | <0.001 | 0.015 | 0.090 | |

* expressed as lactic acid g/100 mL of brine. Each value in the table represents the mean ± standard deviation of two independent assays. [a–c] Means followed by different lowercase letters in each column differed significantly ($p < 0.05$) among trials for a given sampling time. [A–E] Means followed by different capital letters in each row differed significantly ($p < 0.05$) among sampling times for a given trial, as per Tukey's post hoc comparison.

Plan B profiles for salt content, pH, and TTA (Table 2) were similar to those described for plan A; however, a few differences were found, considering that plan B started after the sweetening phase. The pH value at the beginning was slightly lower (4.17) than in plan A, and the same happened at the end (4.2 for AC and ca. 4.75 for A5 and A7). The TTA value at the beginning was almost one third of that under plan A (0.05%), and the same ratio occurred at the end (0.11% for AC and below 0.10% for A5 and A7).

**Table 2.** Evolution in physicochemical features (salt content, pH, and total titratable acidity) of brine at selected times of fermentation under plan B.

| | Trial | Fermentation Time (Days) | | | | |
|---|---|---|---|---|---|---|
| | | **0** | **140** | **182** | **245** | **$p$-Value** |
| Salt (%) | BC | 4.35 ± 0.21 [B] | 3.75 ± 0.07 [aA] | 6.50 ± 0.00 [D] | 5.90 ± 0.14 [C] | <0.001 |
| | B5 | 4.35 ± 0.21 [A] | 4.25 ± 0.07 [bA] | 6.20 ± 0.28 [B] | 5.55 ± 0.07 [B] | 0.001 |
| | B7 | 4.35 ± 0.21 [B] | 3.75 ± 0.07 [aA] | 6.50 ± 0.00 [D] | 5.60 ± 0.00 [C] | <0.001 |
| $p$-value | | 1.000 | 0.009 | 0.253 | 0.057 | |
| pH | BC | 4.17 ± 0.05 [B] | 3.98 ± 0.02 [bA] | 4.01 ± 0.01 [aA] | 4.17 ± 0.04 [aB] | 0.009 |
| | B5 | 4.19 ± 0.01 [B] | 3.76 ± 0.02 [aA] | 4.24 ± 0.02 [bB] | 4.83 ± 0.08 [bC] | <0.001 |
| | B7 | 4.18 ± 0.04 [C] | 3.98 ± 0.02 [bA] | 4.00 ± 0.05 [aAB] | 4.71 ± 0.04 [bD] | <0.001 |
| $p$-value | | 0.859 | 0.003 | 0.019 | 0.003 | |
| TTA * | BC | 0.05 ± 0.00 [A] | 0.21 ± 0.01 [C] | 0.12 ± 0.01 [abB] | 0.11 ± 0.01 [bB] | <0.001 |
| | B5 | 0.05 ± 0.00 [A] | 0.16 ± 0.02 [B] | 0.15 ± 0.01 [bB] | 0.07 ± 0.01 [aA] | 0.003 |
| | B7 | 0.05 ± 0.00 [A] | 0.14 ± 0.02 [B] | 0.07 ± 0.02 [aAB] | 0.07 ± 0.01 [aAB] | 0.038 |
| $p$-value | | 1.000 | 0.130 | 0.017 | 0.032 | |

* expressed as lactic acid g/100 mL of brine. Each value in the table represents the mean ± standard deviation of two independent trials. [a,b] Means followed by different lowercase letters in each column differed significantly ($p < 0.05$) among trials for a given sampling time. [A–D] Means followed by different capital letter in each row differed significantly ($p < 0.05$) among sampling times for a given trial, as per Tukey's post hoc comparison.

Under plan C (Table 3), and after addition of *Lpb. pentosus* i106 to the final product, a decrease in pH (3.6) and an increase in TTA (0.29%) were observed relative to the control (4.04 and 0.14%, respectively, for pH and TTA). As happened under plans A and B, pH and TTA tended to be different (4.46 and 0.07%) by the end of the process, but closer to those of the control (i.e., 4.40 and 0.10%).

**Table 3.** Evolution in physicochemical features (salt content, pH, and total titratable acidity) of brine at selected times of fermentation under plan C.

| | Trial | Fermentation Time (Days) | | | |
|---|---|---|---|---|---|
| | | **0** | **7** | **64** | **$p$-Value** |
| Salt (%) | CC | 5.10 ± 0.14 | 5.75 ± 0.35 | 5.75 ± 0.21 | 0.104 |
| | C5 | 5.00 ± 0.00 [A] | 6.00 ± 0.00 [C] | 5.75 ± 0.07 [B] | <0.001 |
| $p$-value | | 0.423 | 0.423 | 1.000 | |
| pH | CC | 4.03 ± 0.04 [A] | 4.04 ± 0.01 [A] | 4.40 ± 0.08 [B] | 0.011 |
| | C5 | 4.10 ± 0.03 [B] | 3.58 ± 0.01 [A] | 4.46 ± 0.09 [C] | 0.001 |
| $p$-value | | 0.192 | < 0.001 | 0.598 | |
| TTA * | CC | 0.14 ± 0.03 | 0.14 ± 0.02 | 0.10 ± 0.03 | 0.228 |
| | C5 | 0.14 ± 0.01 [B] | 0.29 ± 0.01 [C] | 0.07 ± 0.01 [A] | <0.001 |
| $p$-value | | 0.771 | 0.005 | 0.312 | |

* expressed as lactic acid g/100 mL of brine. Each value in the table represents the mean ± standard deviation of two independent trials. [A–C] Means followed by different capital letter in each row differed significantly ($p < 0.05$) among sampling times for a given trial, as per Tukey's post hoc comparison.

### 3.2.2. Phenolic Composition

Under plan A (Figure 5), no trials differ from each other at the beginning of the experiment. Notice that tyrosol (Figure 5c) is the phenolic compound with the lowest concentration; it underwent little increase with time and reached merely 200–300 mg/L. Trial A0 presented higher tyrosol concentrations than the other trials. Regarding oleuropein concentration (see Figure 5a), all trials, except A0, exhibited similar trends throughout fermentation, i.e., a slight decrease at the beginning of the experiment (from ca. 300 to 50–180 mg/L), followed by a substantial increase (reaching maxima of 300–600 mg/L), and finally a gradual decrease. Inoculated trials (except A0) experienced oleuropein concentrations below the control trial (AC) during most of the fermentation period.

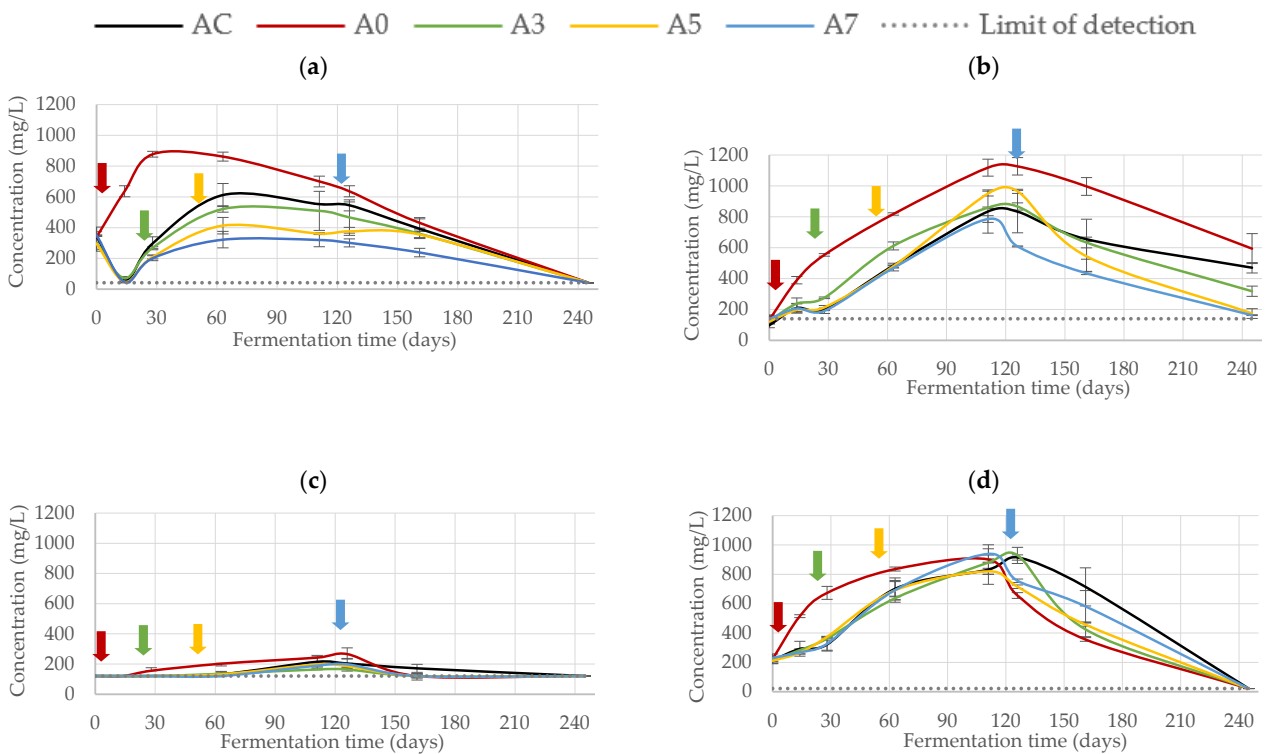

**Figure 5.** Evolution in phenolic profile, (**a**) oleuropein, (**b**) hydroxytyrosol, (**c**) tyrosol, and (**d**) verbascoside, of table olive brine throughout fermentation under plan A. Arrows indicate the moment of inoculation with *Lpb. pentosus* i106 and the corresponding salt addition to all vessels.

Instead of presenting an initial decrease in its oleuropein content (as observed for the other trials), trial A0 exhibited a substantial increase (up to ca. 850 mg/L), with higher values of oleuropein throughout the whole experiment; under higher initial concentrations, the reduction in oleuropein until the end of the experiment was more noticeable than in the remaining trials. Similar to oleuropein, all trials exhibited analogous evolution in hydroxytyrosol concentration throughout the fermentation process (see Figure 5b); it increased from ca. 100 mg/L up to a maximum of 1126 mg/mL by 126 days but underwent a substantial decrease afterwards. Once again, the higher values were associated with trial A0 during the whole experiment; however, by the end of the process, the control trial displayed equivalent higher values as compared to the remaining trials. Moreover, it is possible to identify an apparent correlation between oleuropein content decrease and hydroxytyrosol increase throughout the experiment. Concerning verbascoside (see Figure 5d), the evolution comprised once more an initial increase, beginning at 200 mg/L, until a maximum of 800–1000 mg/L was attained, followed by a substantial decrease; the higher, the larger the salt concentration. The higher value corresponds to the control trial, but the maximum verbascoside concentration was reached later than the previous phenolic compounds.

Under plan B (Figure 6), no differences were found among trials at the beginning of the experiment; no tyrosol was detected either, i.e., concentration was below the detection limit, 121 mg/L. Oleuropein content (Figure 6a) only differed throughout fermentation in trials BC and B7, with an initial increase until a maximum of ca. 140 and 160 mg/mL, respectively, was attained. A few weeks after inoculation, B7 underwent a substantial decrease, while BC remained constant. Regarding trial B5, oleuropein concentration remained low over time (ca. 50 mg/L). Nevertheless, inoculated trials reached final oleuropein concentrations lower than the control trial. The behavior, observed for hydroxytyrosol concentration (Figure 6b), reveals a substantial increase in all trials, followed by a stabilization until the end of the experiment. A higher content was reported in B7, reaching ca. 250 mg/L, while BC remained at ca. 150 mg/L, and B5 reached values very close to the detection limit (140 mg/L). Verbascoside concentration (Figure 6c) also increased with time in all trials, from ca. 50 mg/L to 450 mg/L for BC and B7 and to ca. 300 mg/L for B5. Considering all phenolic compounds, one realizes that the overall concentration of each compound is much lower than the concentration values found under plan A.

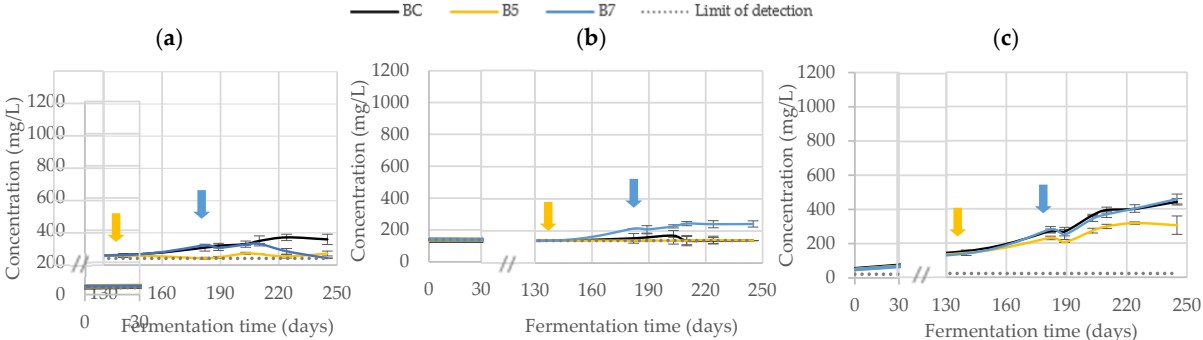

**Figure 6.** Evolution in phenolic profile, (**a**) oleuropein, (**b**) hydroxytyrosol, and (**c**) tyrosol, of table olive brine throughout fermentation under plan B. Arrows indicate the moment of inoculation with *Lpb. pentosus* i106 and the corresponding salt addition to all vessels.

Finally, tyrosol was also not found under plan C (Figure 7). Regarding oleuropein content (Figure 7a), it remained constant throughout the experiment in both trials (CC and C5), at ca. 100–150 and 50–100 mg/L, respectively; one always noticed slightly lower values for trial C5. According to Figure 7b, no major differences were found for hydroxytyrosol content, either through time or within trials; nevertheless, as observed in the previous plans, hydroxytyrosol concentration underwent a little increase until a maximum was reached (from 200 to 400 mg/L) and decreased afterwards to values from 200 to 300 mg/L, being higher for the inoculated trial. Unlike oleuropein content evolution, verbascoside concentration (Figure 7c) started increasing at the very beginning of the experiment, from ca. 50 mg/L to 500 and 250 mg/L for CC and C5, respectively, and remained constant afterwards.

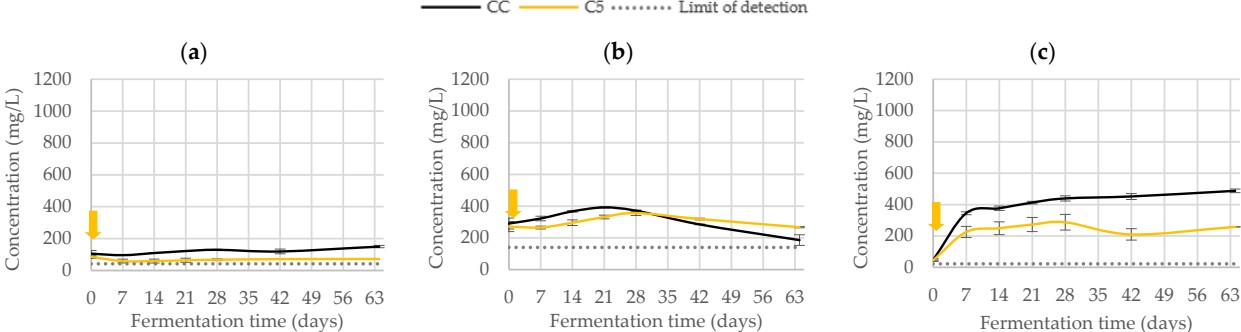

**Figure 7.** Evolution in phenolic profile, (**a**) oleuropein, (**b**) hydroxytyrosol, and (**c**) tyrosol, of table olive brine throughout fermentation under plan C. Arrows indicate the moment of inoculation with *Lpb. pentosus* i106 and the corresponding salt addition to all vessels.

### 3.3. Nutritional Composition

Table 4 exhibits the nutritional profile of the edible part of the olive drupes under the three different plans and by the end of the experiment; this includes moisture, ash, salt, proteins, fats, fiber, and carbohydrate contents, and also the energetic value per each 100 g.

**Table 4.** Nutritional profile of *Cobrançosa* table olives (content per 100 g of edible part of table olives) under the different plans.

| Trial | Content per 100 g of Edible Part of Table Olives | | | | | | | Energetic Value (Kcal/100 g) |
|---|---|---|---|---|---|---|---|---|
| | Humidity | Ashes | Salt | Proteins | Fats | Fiber | Carbohydrates | |
| AC | $66.97 \pm 0.42$ [abc] | $3.22 \pm 0.04$ [a] | $2.60 \pm 0.15$ [a] | $0.92 \pm 0.13$ [ab] | $22.21 \pm 0.49$ [c] | $3.78 \pm 0.11$ [ab] | $1.90 \pm 0.06$ [a] | $218.73 \pm 4.50$ [bc] |
| A0 | $66.28 \pm 0.81$ [a] | $3.29 \pm 0.05$ [ab] | $2.88 \pm 0.01$ [ab] | $1.03 \pm 0.05$ [abc] | $22.82 \pm 0.06$ [c] | $4.46 \pm 0.51$ [ab] | $2.13 \pm 0.25$ [a] | $226.92 \pm 2.36$ [c] |
| A5 | $67.90 \pm 0.29$ [abc] | $3.46 \pm 0.01$ [bc] | $2.95 \pm 0.08$ [b] | $1.16 \pm 0.15$ [abc] | $20.69 \pm 0.48$ [bc] | $4.66 \pm 0.44$ [ab] | $2.14 \pm 0.21$ [a] | $208.68 \pm 2.08$ [b] |
| A7 | $66.36 \pm 0.35$ [a] | $3.49 \pm 0.09$ [c] | $2.94 \pm 0.02$ [b] | $0.89 \pm 0.07$ [a] | $19.81 \pm 1.30$ [b] | $5.84 \pm 1.15$ [ab] | $3.20 \pm 0.67$ [abc] | $207.70 \pm 7.02$ [b] |
| BC | $66.60 \pm 1.22$ [ab] | $3.97 \pm 0.03$ [de] | $3.70 \pm 0.09$ [c] | $1.18 \pm 0.17$ [abc] | $21.08 \pm 0.32$ [bc] | $4.80 \pm 0.50$ [ab] | $2.38 \pm 0.25$ [ab] | $213.54 \pm 1.56$ [bc] |
| B5 | $67.86 \pm 0.23$ [ab] | $3.81 \pm 0.02$ [d] | $3.64 \pm 0.02$ [c] | $1.04 \pm 0.12$ [abc] | $21.34 \pm 0.02$ [bc] | $3.64 \pm 0.25$ [a] | $2.32 \pm 0.15$ [ab] | $212.72 \pm 1.74$ [bc] |
| B7 | $67.12 \pm 1.39$ [ab] | $3.86 \pm 0.06$ [d] | $3.64 \pm 0.08$ [c] | $1.13 \pm 0.02$ [abc] | $20.73 \pm 0.09$ [b] | $4.43 \pm 0.86$ [ab] | $2.73 \pm 0.52$ [abc] | $210.91 \pm 4.55$ [b] |
| CC | $70.22 \pm 0.02$ [c] | $4.07 \pm 0.02$ [e] | $3.61 \pm 0.06$ [c] | $1.34 \pm 0.13$ [c] | $15.34 \pm 0.44$ [a] | $5.98 \pm 0.34$ [b] | $3.47 \pm 0.19$ [b] | $167.89 \pm 2.99$ [a] |
| C5 | $70.02 \pm 0.82$ [c] | $4.12 \pm 0.02$ [e] | $4.02 \pm 0.06$ [d] | $1.47 \pm 0.08$ [c] | $14.64 \pm 0.38$ [a] | $9.41 \pm 0.37$ [c] | $3.78 \pm 0.15$ [c] | $171.58 \pm 5.14$ [a] |
| *p*-value | 0.003 | <0.001 | <0.001 | 0.009 | <0.001 | <0.001 | 0.002 | <0.001 |

Each value in the table represents the mean ± standard deviation of two independent trials. [a–e] Means followed by different lowercase letters in each column differed significantly ($p < 0.05$) among trials, as per Tukey's post hoc comparison.

All nutritional parameters revealed significant differences ($p < 0.05$) between harvesting times (plans A and B from 2021 and plan C from 2020) and some between different protocols of addition of *Lpb. pentosus* i106. Moisture and protein contents were significantly ($p < 0.05$) higher in the 2019/20 harvest (green color) than in its 2020/21 counterpart (color-changing), whereas fat was significantly ($p < 0.05$) lower in 2019/20 than in the 2020/21 harvest. Regarding ash and salt content, the results did differ significantly ($p < 0.05$) between plans: plan C led to the highest values, followed by plan B and, finally, by plan A. Ash content in plan A was higher when the salt percentage added was higher, but no differences were found between plans B and C. Concerning salt content, the control had lower results under plan A conditions, and these were higher in trials A5 and A7; under plan B, no differences were recorded, while the control was significantly ($p < 0.05$) lower under plan C. Fiber content was similar in all trials, except for trials B5 and C5, which exhibited the lowest and highest values, respectively. Regarding carbohydrates, plan A was characterized by lower values, except under trial A7, and plan C was highest. Energetic values also differed between harvests, with significantly ($p < 0.05$) lower values for the 2019/20 harvest.

From the fat content, the relative percentage of fatty acids was determined for the sake of completeness (Table 5). Focusing on the lipidic fraction, the most abundant group of fatty acids was monounsaturated (MUFA), comprising 65–70%, mostly oleic acid (64–68%); saturated fatty acids (SFA) came second (16–18%), being chiefly composed of palmitic (12–14%); next were polyunsaturated fatty acids (PUFA) at 8–12% levels, encompassing mostly linoleic acid (8–11%); finally, *trans* fatty acids appeared at trace quantities. Although statistical differences ($p < 0.05$) were recorded among trials for some of the fatty acids, no pattern was found between harvests, plans, salt addition, or inoculation times.

**Table 5.** Fatty acid profile (% relative of total peak area) of *Cobrançosa* table olives under the different plans.

| Fatty Acid (%Relative) | Trial | | | | | | | | | *p*-Value |
|---|---|---|---|---|---|---|---|---|---|---|
| | AC | A3 | A5 | A7 | BC | B5 | B7 | CC | C5 | |
| Dodecanoic acid (C12:0) | nd | nd | nd | nd | nd | nd | nd | 0.02 ± 0.04 | nd | - |
| Myristic acid (C14:0) | 0.01 ± 0.00 | 0.02 ± 0.00 | 0.02 ± 0.00 | 0.01 ± 0.00 | 0.01 ± 0.00 | 0.01 ± 0.00 | 0.01 ± 0.00 | 0.01 ± 0.00 | 0.01 ± 0.00 | 0.474 |
| Pentadecylic acid (C15:0) | 0.01 ± 0.00 | 0.01 ± 0.00 | 0.01 ± 0.00 | 0.01 ± 0.00 | 0.01 ± 0.00 | 0.01 ± 0.00 | 0.01 ± 0.00 | 0.01 ± 0.00 | 0.01 ± 0.00 | 0.106 |
| Palmitic Acid (C16:0) | 12.53 ± 0.43 [a] | 12.41 ± 0.27 [a] | 12.73 ± 0.28 [a] | 12.72 ± 0.11 [a] | 12.75 ± 0.48 [a] | 12.75 ± 0.38 [a] | 12.64 ± 0.37 [a] | 14.10 ± 0.04 [b] | 13.62 ± 0.43 [b] | <0.001 |
| Heptadecylic acid (C17:0) | 0.15 ± 0.02 | 0.15 ± 0.02 | 0.14 ± 0.01 | 0.14 ± 0.01 | 0.14 ± 0.01 | 0.14 ± 0.01 | 0.14 ± 0.01 | 0.14 ± 0.01 | 0.13 ± 0.01 | 0.185 |
| Stearic acid (C18:0) | 4.20 ± 0.35 [ab] | 3.87 ± 0.17 [ab] | 3.80 ± 0.42 [ab] | 4.25 ± 0.32 [ab] | 4.44 ± 1.08 [b] | 3.63 ± 0.65 [ab] | 3.38 ± 0.39 [ab] | 3.15 ± 0.2 [a] | 3.85 ± 0.25 [ab] | 0.024 |
| Arachidic acid (C20:0) | 0.44 ± 0.04 [bc] | 0.43 ± 0.02 [bc] | 0.45 ± 0.02 [c] | 0.41 ± 0.03 [abc] | 0.41 ± 0.05 [abc] | 0.37 ± 0.04 [ab] | 0.39 ± 0.02 [abc] | 0.35 ± 0.01 [a] | 0.41 ± 0.02 [abc] | 0.002 |
| Heneicosylic acid (C21:0) | 0.04 ± 0.00 [ab] | 0.05 ± 0.02 [ab] | 0.04 ± 0.00 [ab] | 0.04 ± 0.00 [ab] | 0.03 ± 0.00 [a] | 0.04 ± 0.01 [ab] | 0.04 ± 0.01 [ab] | 0.06 ± 0.01 [b] | 0.06 ± 0.01 [b] | 0.006 |
| Behenic acid (C22:0) | 0.09 ± 0.01 [ab] | 0.10 ± 0.01 [ab] | 0.10 ± 0.00 [b] | 0.09 ± 0.01 [ab] | 0.08 ± 0.00 [ab] | 0.08 ± 0.01 [a] | 0.08 ± 0.00 [ab] | 0.08 ± 0.00 [ab] | 0.09 ± 0.00 [ab] | 0.005 |
| Lignoceric acid (C24:0) | 0.06 ± 0.01 [ab] | 0.07 ± 0.00 [ab] | 0.07 ± 0.00 [ab] | 0.06 ± 0.00 [ab] | 0.06 ± 0.00 [ab] | 0.06 ± 0.00 [ab] | 0.05 ± 0.01 [a] | 0.06 ± 0.00 [ab] | 0.07 ± 0.00 [b] | 0.021 |
| ∑ SFA | 17.49 ± 0.50 [abc] | 17.09 ± 0.39 [ab] | 17.31 ± 0.32 [abc] | 17.69 ± 0.34 [bc] | 17.90 ± 0.66 [bc] | 17.06 ± 0.43 [ab] | 16.70 ± 0.17 [a] | 17.93 ± 0.23 [bc] | 18.19 ± 0.39 [c] | <0.001 |
| Palmitoleic acid (cis-C16:1) | 0.93 ± 0.18 [a] | 0.98 ± 0.05 [a] | 0.95 ± 0.02 [a] | 1.05 ± 0.04 [ab] | 0.94 ± 0.12 [a] | 1.09 ± 0.21 [ab] | 1.05 ± 0.04 [ab] | 1.37 ± 0.09 [bc] | 1.27 ± 0.02 [c] | <0.001 |
| Ginkgolic acid (C17:1) | 0.26 ± 0.04 | 0.25 ± 0.02 | 0.24 ± 0.02 | 0.24 ± 0.03 | 0.22 ± 0.03 | 0.25 ± 0.02 | 0.25 ± 0.02 | 0.25 ± 0.02 | 0.22 ± 0.01 | 0.336 |
| Oleic acid (cis-C18:1) | 65.52 ± 0.82 [ab] | 67.10 ± 1.66 [bc] | 66.72 ± 0.84 [abc] | 64.80 ± 1.68 [ab] | 64.94 ± 1.66 [ab] | 67.06 ± 0.94 [bc] | 68.51 ± 0.52 [c] | 65.79 ± 0.15 [ab] | 64.19 ± 0.82 [a] | <0.001 |
| Eicosaenoic acid (C20:1) | 0.22 ± 0.02 [b] | 0.21 ± 0.01 [ab] | 0.22 ± 0.00 [b] | 0.21 ± 0.01 [ab] | 0.21 ± 0.01 [ab] | 0.21 ± 0.01 [ab] | 0.21 ± 0.01 [ab] | 0.18 ± 0.01 [a] | 0.20 ± 0.02 [ab] | 0.009 |
| Docosaenoic acid (C22:1) | 0.02 ± 0.00 | 0.02 ± 0.00 | 0.02 ± 0.00 | 0.02 ± 0.00 | 0.01 ± 0.01 | 0.02 ± 0.00 | 0.01 ± 0.01 | 0.02 ± 0.02 | 0.02 ± 0.00 | 0.729 |
| ∑ MUFA | 66.95 ± 0.83 [a] | 68.56 ± 1.63 [ab] | 68.15 ± 0.83 [ab] | 66.32 ± 1.73 [a] | 66.32 ± 1.78 [a] | 68.62 ± 1.11 [ab] | 70.02 ± 0.51 [b] | 67.62 ± 0.09 [a] | 65.90 ± 0.84 [a] | <0.001 |
| Linoleic acid (C18:2) | 10.17 ± 0.88 [bc] | 8.24 ± 0.66 [a] | 9.12 ± 0.72 [ab] | 10.19 ± 0.56 [bc] | 10.72 ± 0.87 [c] | 9.20 ± 0.82 [abc] | 7.93 ± 0.27 [a] | 8.79 ± 0.34 [ab] | 10.12 ± 0.50 [bc] | <0.001 |
| α-Linolenic acid (C18:3) | 0.86 ± 0.06 [ab] | 0.81 ± 0.02 [a] | 0.87 ± 0.03 [ab] | 0.85 ± 0.10 [ab] | 0.76 ± 0.07 [a] | 0.74 ± 0.06 [a] | 0.84 ± 0.03 [ab] | 0.84 ± 0.02 [ab] | 0.95 ± 0.05 [b] | <0.001 |
| ∑ PUFA | 11.03 ± 0.91 [bc] | 9.04 ± 0.66 [a] | 9.99 ± 0.72 [abc] | 11.04 ± 0.57 [bc] | 11.49 ± 0.93 [c] | 9.94 ± 0.83 [abc] | 8.77 ± 0.29 [a] | 9.63 ± 0.34 [ab] | 11.07 ± 0.54 [bc] | <0.001 |
| Palmitelaidic acid (trans-C16:1) | nd | nd | nd | nd | nd | nd | nd | nd | nd | - |
| Elaidic acid (trans-C18:1) | nd | 0.01 ± 0.01 | 0.01 ± 0.00 | 0.01 ± 0.01 | 0.01 ± 0.01 | nd | 0.01 ± 0.01 | 0.01 ± 0.01 | 0.01 ± 0.01 | 0.957 |
| Linoelaidic acid (trans-C18:2) | 0.01 ± 0.00 | 0.01 ± 0.00 | 0.01 ± 0.00 | 0.02 ± 0.01 | 0.01 ± 0.00 | 0.02 ± 0.01 | 0.01 ± 0.00 | 0.02 ± 0.02 | 0.01 ± 0.00 | 0.382 |
| ∑ trans | 0.02 ± 0.00 | 0.02 ± 0.01 | 0.02 ± 0.01 | 0.03 ± 0.01 | 0.02 ± 0.01 | 0.02 ± 0.00 | 0.02 ± 0.00 | 0.03 ± 0.02 | 0.02 ± 0.01 | 0.779 |

Each value in the table represents the mean ± standard deviation of two independent trials. [a–c] Means followed by different lowercase letters in each column differed significantly (*p* < 0.05) among trials, as per Tukey's post hoc comparison.

### 3.4. Sensory Assessment

Figure 8 pertains to the sensory analysis under plan A (Figure 8a), plan B (Figure 8b), and plan C (Figure 8c) and encompasses table olive texture, flavor, saltiness, bitterness, acidity, and abnormal flavors (typically "cooked" or "metallic" ones). The five trials under plan A did not reveal sensory differences for salty (ca. 4.5) or abnormal flavors (ca. 4.5 points). Differences of texture between trials were found; it was better for trials with early inoculation (ca. 4) and then decreased to 3 or 2 for A5 and A7, respectively. Flavor quality was worse for trial A7 (4), when compared to the other trials, which proved essentially equivalent to each other (6). The bitterness of drupes when the strain was added early in the processing period under plan A was higher than that when the strain was added later on; it was 6 for AC and A0, 5 for A3, 4 for A5, and 3 for A7. A similar pattern was found for the acidity of table olives, i.e., 5 for AC and A0, 4 for A3, and 3 for A5 and A7. Regarding plan B, in addition to salty (ca. 5) and abnormal flavors (ca. 4.5 points), the texture did not exhibit major differences (ca. 5). The quality of the flavor was better, and equal in BC and B5 (ca. 6.0), when compared to B7 (4.5). Bitterness decreased to ca. 3 upon addition of *Lpb. pentosus* i106, both in B5 and B7, with a value of 5 for the control. Acidity also decreased (ca. 3.5) with the addition of strain i106 when compared to control (ca. 5.5). No major differences between the two trials under plan C could be found for the six sensory parameters. Figure 8d allows comparison of the overall eating quality (OEQ) between plans. One found that points given to OEQ under plan C (green olives) were statistically higher than those found under plans A and B (both with color-changing olives). From Figure 8d, one finds better evaluations under plan B (i.e., when *Lpb. pentosus* i106 was added after sweetening) than under plan A. Under plan B, the highest value for OEQ was obtained for B7; however, the opposite was observed under plan A, i.e., a lower OEQ was observed for the last trial (A7).

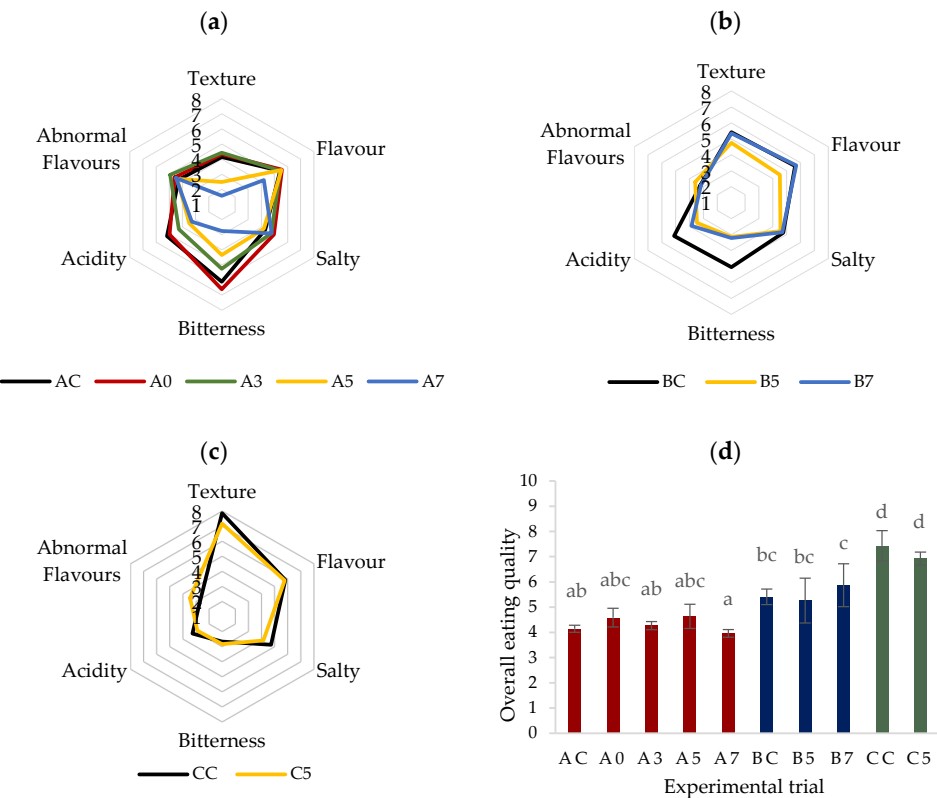

**Figure 8.** Sensory analysis of *Cobrançosa* table olives fermented under (**a**) plan A, (**b**) plan B, and (**c**) plan C and (**d**) overall eating quality under the different plans. Each column represents the mean ± standard deviation of two independent trials. [a–d] Means followed by different lowercase letters in each column differed significantly ($p < 0.05$) among trials, as per Tukey's post hoc comparison.

## 4. Discussion

The maturation degree of table olives has a considerable impact upon nutritional composition; therefore, plan C should be discussed separately from plans A and B. In order to discuss all results obtained as a whole while focusing on the main objective of this study, i.e., production of *Cobrançosa* table olives with a potentially probiotic character via addition of an endogenous LAB culture, one should look for the plans that produced a better sensory evaluation and (if possible) for the best results attained under each plan. Under such circumstances, it will be easier to justify the profiles of the other parameters analyzed.

Besides varying, in general, with cultivar or processing method, the nutritional properties of table olives depend on some agriculture practices, preharvest factors (such as irrigation and fruit ripening stage), agroclimatic conditions, and geographical origin [22,26]. In this study, it was possible to control cultivar and fruit ripening stage (which relate to the harvest decision). Regarding cultivar, previous studies on cv. *Cobrançosa* found nutritional values very similar to those in this study [27,28]. However, the color of the olive, which is directly related to the maturation degree of the drupe, plays a role in the nutritional value of the final product. As explained by Rocha et al. [26], since the maturation of olives occurs gradually, the fat content of the olive flesh, and consequently the energetic value, increase simultaneously along with a decrease in water content; this corroborates the results of this study. Fiber content also differed between maturation degrees, being higher in the 2019/20 harvest olives (green olives); this happens because the activity against the cell wall leads to fiber degradation, namely of cellulose and hemicellulose, during maturation [29]. Focusing on the lipid fraction, the levels of oleic (64–68%), palmitic (12–14%), and linoleic (8–11%) acid residues mimic in general olive-based products [22]. Once again, the degree of ripeness of olives has been reported to affect the nutritional profile; in this case, this includes the fatty acid profile. MUFA did indeed decrease, while PUFA increased with maturity [22,30]; other studies showed that both MUFA and PUFA increase with ripening progress [31], which is in agreement with the present study. Such observations might be due to the fact that fatty acids respond in general more heterogeneously to stress and subsequent recovery [32]. It must be kept in mind that said fatty acid composition is associated with nutritional and health attributes, mainly owing to the richness in MUFAs, that are recognized to reduce cardiovascular diseases and to decrease levels of SFAs, which are related to an increased content of LDL cholesterol [33].

Based on overall quality sensory evaluation, the best plan was plan C. The first reason for this is probably the shorter maturation time of the table olives; a similar preference for green olives over color-changing olives was reported by Hurtado et al. [34] for *Arbequina* natural table olives. Texture was one of the parameters differing among plans; this can be explained by the hydrolysis of cell wall pectic polysaccharides during fermentation, which results in the loss of structural coherence of olive tissues, eventually leading to softer textures associated with negative sensory appreciation. Furthermore, salt added too soon is more likely to soften the olives, as per the induced osmotic dehydration process that gradually damages cell tissues [35]. Second, it is possible to infer that the sensory quality of the final product is not affected when *Lpb. pentosus* i106 is added to green table olives as an additive at the packaging stage; this is apparent from the similarities between trials at their final stage in terms of physicochemical and debittering processes (oleuropein depletion), considering that the LAB strain grows substantially during the first week and remains stable in number thereafter but also starts to grow after 40 days in the control trial. Focusing on the differences between plans A and B, one may infer that the existence of a sweetening stage had a positive impact upon the overall quality of the final color-changing table olives; therefore, this stage should be maintained within the fermentation process for its lack of interference with the addition of a potential probiotic LAB culture.

Under plan B, the control and the two protocols exhibited the same levels of texture and salt, as expected due to the similar maturation degrees of table olives and times of salt addition, as well as abnormal flavors; these consubstantiate good features. The control appears to be preferred, mainly because of higher acidity and bitterness; these two

sensory parameters can thus be claimed as possessing a higher relevance for the evaluation of the sensory properties of this particular cultivar. The sensory analysis agrees with results obtained for pH, TTA, and phenolic compounds; trials leading to higher acidity and bitterness exhibited in fact lower pH, higher TTA, and higher phenolic content, namely oleuropein. Although the addition of *Lpb. pentosus* i106 produces an increase in LAB numbers in both brine and drupes and consequently pH and biochemical profiles, the spontaneous growth of LAB compensates once again for these variations at a later stage of the process in the control in such a way that the final pH and biochemical concentrations of the control contribute to a better overall sensory quality, mainly due to the lower pH and the higher TTA and oleuropein levels. These two findings can be rationalized by the fact that, in trials B5 and B7, *Lpb. pentosus* i106 helped accelerate the process of: (i) the production of lactic acid, which is then converted to other organic acids with the aid of yeasts and (ii) oleuropein degradation [11]. Finally, the scores for flavor were similar between control and trial B5.

In the case where the sweetening stage is absent (plan A), no significant ($p < 0.05$) differences are obtained in terms of overall eating quality; hence, the best moment to add the strain is at the beginning (trial A0), based on the similarity of all sensory parameters to those of the control and the poorer sensory assessment when strain addition occurs later in the process; in this case, a true probiotic starter culture would be at stake.

Microbial evolution revealed that microorganism populations, either LAB or yeasts, are at higher levels in brine than in drupes, with differences of ca. 1 log, irrespective of the year of harvest; Anagnostopoulos et al. [23] reported similar differences. This finding is probably related to the fact that LAB and yeasts are typically found in pores, lesions, lenticels, or irregularities in the olive surface [11,36].

The dominant microbiota in control trials throughout the process were yeasts; LAB were only detected at advanced stages of salting, as expected in view of previous findings by Reis et al. [8]. However, this study concluded that yeast populations keep their growth throughout the process, irrespective of an increasing salt content or the addition of higher numbers of *Lpb. pentosus* i106 (inoculum up to 8 log CFU/mL in brine); on the other hand, LAB take longer to adapt to the new environmental conditions when the percentage of salt increases, as apparent under plans A and B. No matter the plan followed, potential probiotic LAB strain i106 is present by the end of the process in high numbers in brine (ca. 7–8 log CFU/mL) and drupes (ca. 6–7 log CFU/g), as intended. The results of plan B help conclude that the optimum salt content is 5–6%, should the purpose be having probiotic table olives throughout fermentation. This agrees with Penland et al. [37], who showed that LAB populations were below detection limits in all fermentations due to the high salt content in brines (8–10%). Tassou et al. [38] also claimed that 4 and 6% (w/w) NaCl brines favored LAB growth when compared to 8% (*w/w*) NaCl brines. Plan C emphasizes that the potential probiotic LAB i106 was viable by 1 month of storage, which guarantees the probiotic potential of table olives during (at least) that period; nevertheless, this is a short time for typical commercial storage, namely for producers and wholesalers, meaning that further monitoring will be necessary over time to ascertain strain viability and stability in the longer run.

Spoilage or even pathogenic species may grow during the first stages of fermentation, but they usually decay rapidly compared to the growth of yeasts and LAB, since they are more sensitive to salt concentration and the acidification of brines brought about by the metabolic activity of LAB [4]; this is consistent with results generated in this study. Under plan A, before any sweetening or salting, either in brine or in drupes, those populations were substantially higher at the beginning of the experiment and contracted upon addition of salt and strain *Lpb. pentosus* i106 in each trial to reach values below the detection limit, as long as pH decreases and salt and titratable acidity increase. In this case, it seems that the addition of potentially probiotic LAB i106 helped reduce Enterobacteriaceae faster than in the control (at lower salt contents) due to its own co-aggregation with such pathogens as *Escherichia coli* ATC25922, as reported previously by Coimbra-Gomes et al. [16]. Considering salt content

and pH values at the initial period under plans B and C and the corresponding values under plan A for the same time, Enterobacteriaceae were not expected, as effectively observed in this study. Servili et al. [39] also observed this type of decrease in Enterobacteriaceae populations as *Lpb. pentosus* populations increased in black table olives (*cv. Itrana* and *Leccino*). Regarding the three groups of microorganisms, Anagnostopoulos et al. [40] reported quite similar microbial evolutions along the fermentation of natural whole and cracked *Picual* table olives, inoculated with a *Lpb. plantarum* strain; they also concluded that inoculation accelerates and favors the elimination of unwanted microorganisms.

As stressed before, yeasts are not affected during *Cobrançosa* fermentation, yet the time of salt addition and its concentration affect growth rate and biomass reached by the potentially probiotic LAB strain i106; this is reflected by the physicochemical features and the extent of degradation of oleuropein when compared to the control.

The pH profiles found were consistent with *Lpb. pentosus* i106 growth. At each time of sampling, trials characterized by earlier inoculation with the potentially probiotic LAB culture always exhibited lower pH and higher titratable acidity. Irrespective of the plans, the pH profiles for trials A0 and A3 were similar, as expected from the tolerance to pH and similar growth rate of these strains below 5% salt (unpublished data). Penland et al. [37] reported a similar pH tendency in *Nyons* black table olives, beginning at 5.4, followed by a decrease to 4.4, and remaining constant afterwards. On the other hand, the profiles of pH for trials A5 and A7 looked the opposite; lower tolerance to higher salt contents led to poorer adaptation of the strain to the environment, thus causing a lower growth rate, which resulted in higher pH values and lower acidification. This was also found by Anagnostopoulos et al. [23], with LAB growth apparently hampered by salt-tolerant yeast species, thus resulting in a less acidic product.

In parallel, titratable acidity increased throughout the first days of fermentation, from 0.15–0.56, 0.05–0.21, and 0.14–0.29 g/100 g lactic acid under plans A, B, and C, respectively, namely for those trials undergoing earlier inoculation. This is mainly due to: (i) LAB activity in the synthesis of organic acids, such as lactic acid; (ii) diffusion and solubilization of those compounds from the table olive tissues; and (iii) increase in free fatty acids present in the drupes or produced by microorganisms [41]. Similar results were reported by Lanza et al. [13], i.e., 0.17–0.30 g/100 g lactic acid. By the end of the experiment, however, titratable acidity suffered a significant ($p < 0.05$) decrease in all trials. Acidity decrease (or pH increase) is justified by yeast dominance over LAB populations, which metabolize organic acids, as reported by Anagnostopoulos et al. [40], Aponte et al. [42], Panagou et al. [43], Papadelli et al. [44], and Reis et al. [8]; in fact, under plans B and C, characterized by a lower titratable acidity, yeast populations exist to quite similar numbers as compared to LAB populations.

As mentioned before, green table olives received better sensory scores than the color-changing ones; it was also emphasized that bitterness is an intrinsic requirement for good quality *Cobrançosa* table olives. This agrees with the fact that, during the growth phase of olives, the total phenol concentration (in particular oleuropein) increases to a maximum level at the green maturation phase, followed by a substantial reduction; however, there is continuous synthesis of these compounds until maturity [26]. In agreement with this study, Sousa et al. [45] identified oleuropein as the main phenolic compound at the first stages of maturation, which substantially decreased during ripening, while hydroxytyrosol became predominant at intermediate and long maturation stages. Although oleuropein is the major component responsible for bitter taste in olive fruits, it tends to disappear throughout fermentation due to bacterial activity or endogenous and chemical hydrolysis, e.g., by β-glucosidase enzymatic activity, thus leading to less bitter phenolic compounds (e.g., hydroxytyrosol); this makes olives edible and thus increases acceptance by the consumer [26,37]. This point was confirmed during some periods of fermentation but was clearer in the trials where potentially probiotic *Lpb. pentosus* i106 culture was inoculated earlier. Considering all phenolic compounds analyzed under all plans, the concentration evolution throughout fermentation follows trends reported elsewhere [8]. Furthermore, the

LAB strain i106 has been previously proven to hold the ability to degrade oleuropein (unpublished data). A similar LAB debittering activity during fermentation was reported by Lanza et al. [13]. Oscillations in phenolic compounds under plan A are also highly affected by the washing and sweetening processes; as observed in Figure 5a, a substantial decrease in oleuropein content by two weeks is observed in all trials but A0. This is justified by the lack of washing of said trial prior to inoculation. Although the goal here was to reproduce an actual starter culture, this procedure led to higher phenolic compound concentrations throughout the whole experiment for this trial, as outlined in Figure 5. The importance of the sweetening process is once again highlighted by the much lower overall phenolic concentrations found under plans B and C when compared to plan A, since both were submitted to sweetening, whereas plan A was not; by washing the olives, the phenolics that have diffused away from the flesh to the brine are necessarily eliminated [8].

## 5. Conclusions

The maturation degree of table olives plays an important role in the overall eating quality of this type of cultivar, especially in terms of texture; drupes should accordingly be processed at the final stage of green before turning color, which calls for the definition of an index of maturation, aiming at future process standardization. Reduction in salt content to 5% in the final product is also recommended, not only to reinforce growth and extend viability of potentially probiotic *Lpb. pentosus* i106, but also to enhance health benefits associated with low salt ingestion. To get the best sensory scores for *Cobrançosa* table olives, an earlier sweetening stage must be present as part of their processing, as it is meant to remove excess phenolic compounds and bring down the degree of bitterness to acceptable levels. However, a potentially probiotic starter culture comprising strain *Lpb. pentosus* i106 may not function in loco for this type of process because of the discarding of brine already inoculated, meaning that such a strain is to act as a supplementary additive (or adjunct culture). As a consequence, the best plan found was the addition of the potentially probiotic culture to the ready-to-eat table olives, considering that it was possible to obtain a final product with overall eating quality similar to that of the control (obtained via plain fermentation by ill-defined, adventitious strains).

## 6. Future Research

Considering that yeasts were the dominant microorganisms over the whole fermentation process, future studies should also resort to isolation and characterization of yeast strains native in *Cobrançosa* cultivar in attempts to develop conditions suitable for yeast probiotic strains, or even new starters, by combining both LAB and yeasts, thus mimicking the adventitious microbiota of olives. Further technological studies could also look forward to new fermentation strategies that favor the survival and dominance of other LAB probiotic starter strains, already characterized within our research group; this includes (but is not limited to) the use of post-probiotics, of biofilms, or of microencapsulated microorganisms, considering that one major conclusion of this study points to the addition of the culture during final packaging.

**Author Contributions:** J.C.-G.: Data curation, Formal Analysis, Investigation, Visualization, Writing—original draft and Writing—review and editing; P.J.M.R.: Conceptualization, Data curation, Investigation, Methodology, Supervision, Validation and Writing—review and editing; T.G.T.: Investigation and Methodology; A.A.S.: Investigation; E.M.: Investigation; S.C.: Investigation; F.X.M.: Funding acquisition, Project administration, Supervision, Validation and Writing—review and editing; A.C.M.: Conceptualization, Formal analysis, Funding acquisition, Investigation, Methodology, Project administration, Supervision, Validation and Writing—review and editing. All authors have read and agreed to the published version of the manuscript.

**Funding:** This work was financially supported by: Project PROMETHEUS-POCI-01-0145-FEDER-029284, funded by FEDER funds through COMPETE2020—Programa Operacional Competitivi-dade e Internacionalização (POCI) and by national funds (PIDDAC) through FCT/MCTES; LA/P/0045/2020 (ALiCE), UIDB/00511/2020, and UIDP/00511/2020 (LEPABE), funded by national funds through

FCT/MCTES (PIDDAC); and UIDB/50006/2020 and UIDP/50006/2020 (LAQV/REQUIMTE), funded by national funds through FCT/MCTES. Joana Coimbra-Gomes and Afonso A. Silva were financially supported by Research Grants under the Partnership Agreement with the Faculty of Engineering of the University of Porto, inserted in project PROMETHEUS-POCI-01-0145-FEDER-029284. Author Patrícia J.M. Reis was financially supported by a work contract, inserted in project PROMETHEUS-POCI-01-0145-FEDER-029284.

**Institutional Review Board Statement:** Not applicable.

**Informed Consent Statement:** Not applicable.

**Data Availability Statement:** Data are contained within the article.

**Acknowledgments:** Eng. Bruno Costa, from a Mirandela local SME, is acknowledged for providing samples of *Cobrançosa* table olives for this work and for his time and availability for the logistic arrangements underlying sample supply; we also acknowledge all panelists who filled in the sensory survey forms for this research project.

**Conflicts of Interest:** The authors declare no conflict of interest. The funders had no role in the design of the study; in collection, analyses, or interpretation of data; in writing of the manuscript; or in decision to publish the results.

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
