# Peer review of "Cobrançosa Table Olive Fermentation as per the Portuguese Traditional Method, Using Potentially Probiotic Lactiplantibacillus pentosus i106 upon Alternative Inoculation Strategies"

_fermentation, doi:10.3390/fermentation9010012_

Round 1
Reviewer 1 Report
The work is well written, and the data is properly presented, but I have two main concerns.
First, the experimental plan needs to be detailed. There are several missing explanations. For example, why did the trial C ready-to-eat olive monitoring last only one month? And how is it conceivable that naturally occurring LAB in this trial were undetectable prior to the starter's inoculation?
Additionally, why did salt addition constantly concomitant with the strain's inoculation? Figure 1 generally enhances comprehension, but it is necessary to explain the reasons for the authors’ decisions.
The use of the term "probiotic" is the subject of my second objection. The two investigations that support this status for the adopted strain are insufficient, especially given that neither study includes a clinical trial. So, the word' probiotic' should be constantly preceded by 'potential' and should disappear from the title.
With reference to the material and methods, I believe that media and conditions for microbial monitoring should be reported.
Conclusions are not understandable and need to be rewritten.
Author Response
Point 1: First, the experimental plan needs to be detailed. There are several missing explanations.
For example, why did the trial C ready-to-eat olive monitoring last only one month?
Response 1: In the original plan, trial i106 at the final stage had not been included. However, as the experimentation evolved, we realized that plan C was essential to better understand all stages (i.e. sweetening, salting, ready-to-eat). The reason why it lasted only one month was because of the narrow availability to meet of the taste panel members; and because addition of said probiotic after packaging would compromise its viability (as a post biotic, as further discussed in the revised version as per suggestion by another reviewer).
Point 2: And how is it conceivable that naturally occurring LAB in this trial were undetectable prior to the starter's inoculation?
Response 2: In Introduction section, we briefly described the manufacture process of Cobrançosa table olive (see ref 9); according to said paper, we conclude that the variability of LAB among producers is quite high, and that the olives from some producers did not actually exhibit presence of significant numbers of (viable) LAB until the salting stage. Despite being a somewhat awkward realization, it actually reflects experimental evidence.
Point 3: Additionally, why did salt addition constantly concomitant with the strain's inoculation? Figure 1 generally enhances comprehension, but it is necessary to explain the reasons for the authors’ decisions.
Response 3: As mentioned before, the manufacture of Cobrançosa table olives is briefly described in our manuscript, yet characterized in detail in paper ref. 9. The variability among producers is indeed quite large. Therefore, we tried to find the best compromise between the results and conclusions reported in previous work (paper ref 9) in the current manuscript. For us, it made sense to add the (potentially) probiotic strain at the same time as the salt concentration is increased; in any case, some additional explanation was added in the revised manuscript to explain this option better.
Point 4: The use of the term "probiotic" is the subject of my second objection. The two investigations that support this status for the adopted strain are insufficient, especially given that neither study includes a clinical trial. So, the word' probiotic' should be constantly preceded by 'potential' and should disappear from the title.
Response 4: In agreement with your suggestion, we added the term “potential” to probiotic in the whole text; by the same token, we believe it is important to keep the “potentially probiotic” term in the title as well.
Point 5: With reference to the material and methods, I believe that media and conditions for microbial monitoring should be reported.
Response 5: Such piece of information was added to this section, as suggested.
Point 6: Conclusions are not understandable and need to be rewritten.
Response 6: Conclusions were carefully re-written, as suggested.

Reviewer 2 Report
The paper treats an interesting topic referring to production of Cobrançosa table olives using an endogenous (probiotic) Lactiplantibacillus strain, previously isolated from Cobrançosa table olives and brines.
There are some adjustments that need to be made:
1. In Abstract the amount of NaCl added to olives is expressed in (w/v) for plan A, but for samples B and C this information is missing.
2. All acronyms should be explained at their first use (i.e. LAB in Abstract, OD in Materials and Methods section).
3. The equipment used for measuring the optical density of aliquots is not mentioned.
4. Please check the information from rows 137 and 139: the addition of LAB or the addition of Lpb. pentosus i106?
5. Please explain the information stated in rows 512 - 513 and 519, referring to the presence and viability of the strain Lpb. pentosus i106 in brine and drupes. In correlation with the data presented in Figures 2 - 4, including the evolution of LAB in control samples, taking into account that the experimental design is not identical with that one mentioned by Reis, et al. [9], how it can be stated with certainty that the LAB population consists only of Lpb. pentosus i106?
Extensive explanations and correlation between the analyzed parameters are provided by authors, underlying the practical value of research. Further perspective of research could be mentioned.
Author Response
Point 1: In Abstract the amount of NaCl added to olives is expressed in (w/v) for plan A, but for samples B and C this information is missing.
Response 1: The piece of information referred to was added, as suggested.
Point 2: All acronyms should be explained at their first use (i.e. LAB in Abstract, OD in Materials and Methods section).
Response 2: In the Abstract, we replaced “LAB” by “lactic acid bacteria”. OD stands for Optical Density, and was so included in full in the first occurrence in Materials and Methods section.
Point 3: The equipment used for measuring the optical density of aliquots is not mentioned.
Response 3: Sid piece of information was added, as suggested.
Point 4: Please check the information from rows 137 and 139: the addition of LAB or the addition of Lpb. pentosus i106?
Response 4: In fact, we meant “The addition of Lpb. pentosus i106”; the text was corrected accordingly.
Point 5: Please explain the information stated in rows 512 - 513 and 519, referring to the presence and viability of the strain Lpb. pentosus i106 in brine and drupes. In correlation with the data presented in Figures 2 - 4, including the evolution of LAB in control samples, taking into account that the experimental design is not identical with that one mentioned by Reis, et al. [9], how it can be stated with certainty that the LAB population consists only of Lpb. pentosus i106?
Response 5: We did not state in the main text that LAB population consists only of Lpb. pentosus i106. In fact, we are aware that this strain is present to high numbers in brines and drupes, because the culture was pure at the moment of inoculation (as also confirmed via optical microscopy); and every step was handled under sterile conditions (space, chemicals, and tools). Therefore, if an increase in population after inoculation of our culture were detected, then such increase would certainly be due (only) to our strain (which, as mentioned above, was pure).
Point 6: Extensive explanations and correlation between the analyzed parameters are provided by authors, underlying the practical value of research. Further perspective of research could be mentioned.
Response 6: Some further perspective of research was added to the main text, as suggested.

Reviewer 3 Report
Manuscript: Fermentation 2064688
The manuscript deals with adding a starter culture of Lactiplantibacillus pentosus i106 in different conditions to lead to a more standardized process and add commercial value to the olives. To this aim, several physicochemical and nutritional parameters were followed. The manuscript includes numerous determinations and a sensory and nutritional evaluation analysis over time. However, some questions should be considered.
Title. The preparation of table olives is diverse. The title should mention the type or style the manuscript is referred to.
L20 According?
L21. What is regular sweetening? Readers are probably non-specialists in the matter. Explain the meaning.
L22. It sounds rare that the first season was the last in the explanation. It is certain that the 2019/20 experiment was performed first and, possibly, somehow induced the following ones. Or not?
L30. The sentence is obscure.
L39-40. The reference is inadequate.
L42. This reference is also inadequate. Please revise the use of references in the text. They should be related to the issues in context.
L65-66. The period in water is long. What could be the safety risk level?
L65-67 renewals of brines? When the salt or brine was added?
L65-70. The process should be explained better.
L129 How were the olives preserved till elaboration?
L131-147. The experimental design is hard to follow. Furthermore, Fig 1 is also complex, and its contribution to understanding the experiment is limited. One may understand that processing is rather complicated, but a clear explanation is necessary to follow the evolution properly.
L178. A reference could be pertinent.
Figure 2-4. Identification of lines should be situated on the top of the plots. Its current situation may induce error. Besides, the meaning of the arrow colours is not explained.
Tables 1-5. The use of average± standard deviation in tables has an uncertain meaning. From the statistical side, it only represents about 68% population. Besides, their values for the same parameters on different samples are diverse. They may cause doubts about such differences' origin (method, matrixes, …). Since the authors applied ANOVA for comparison, it would be more informative to remove the individual values in each average for a pooled standard deviation at the end of each column just above the p-value. Maybe including the mean value for each parameter and the standard deviation (in parenthesis) could also facilitate identifying more/less favourable treatments more straightforwardly. Furthermore, the p-value could be removed since the footnote includes the limit for significance. Eliminating standard values in each row would allow a more straightforward comparison among means.
Comments to Figures and Table. Please revise the comparisons and hypothesis. Various treatments share the same letters, but many have only one. Then identifying differences could be not straightforward. Also, the hypothesis based on parameter values could be misleading (e.g. L383 fat content tended to decrease with time of addition of probiotic strain and salt concentration; was the fat used by the organisms?, How could the salt destroy the fat?; L405, Bitterness increased as the strain is added earlier… ???). Also, there are statements excessively vague (e.g., L410, bitterness decreased. .., but statistical support is not provided).
L610. Legal limits should also be considered.
Author Response
We do not received reviewer's comments.

Round 2
Reviewer 3 Report
Manuscript: Fermentation 2064688
The manuscript deals with adding a starter culture of Lactiplantibacillus pentosus i106 in different conditions to lead to a more standardized process and add commercial value to the olives. To this aim, several physicochemical and nutritional parameters were followed. The manuscript includes numerous determinations over time and a sensory and nutritional evaluation analysis. However, some questions should be considered.
Title. The preparation of table olives is diverse. The title should mention the type or style the manuscript is referred to.
L20 According?
L21. What is regular sweetening? Readers are probably non-specialists in the matter. Explain the meaning.
L22. It sounds rare that the first season was the last in the explanation. It is certain that the 2019/20 experiment was performed first and, possibly, somehow induced the following ones. Or not?
L30. The sentence is obscure.
L39-40. The reference is inadequate.
L42. This reference is also inadequate. Please revise the use of references in the text. They should be related to the issues in context.
L65-66. The period in water is long. What could be the safety risk level?
L65-67 renewals of brines? When the salt or brine was added?
L65-70. The process should be explained better.
L129 How were the olives preserved till elaboration?
L131-147. The experimental design is hard to follow. Furthermore, Fig 1 is also complex, and its contribution to understanding the experiment is limited. One may understand that processing is rather complicated, but a clear explanation is necessary to follow the evolution properly.
L178. A reference could be pertinent.
Figure 2-4. Identification of lines should be situated on the top of the plots. Its current situation may induce error. Besides, the meaning of the arrow colours is not explained.
Tables 1-5. The use of average± standard deviation in tables has an uncertain meaning. From the statistical side, it only represents about 68% population. Besides, their values for the same parameters on different samples are diverse. They may cause doubts about such differences' origin (method, matrixes, …). Since the authors applied ANOVA for comparison, it would be more informative to remove the individual values in each average for a pooled standard deviation at the end of each column just above the p-value. Maybe including the mean value for each parameter and the standard deviation (in parenthesis) could also facilitate identifying more/less favourable treatments more straightforwardly. Furthermore, the p-value could be removed since the footnote includes the limit for significance. Eliminating standard values in each row would allow a more straightforward comparison among means.
Comments to Figures and Table. Please revise the comparisons and hypothesis. Various treatments share the same letters, but many have only one. Then identifying differences could be not straightforward. Also, the hypothesis based on parameter values could be misleading (e.g. L383 fat content tended to decrease with time of addition of probiotic strain and salt concentration; was the fat used by the organisms?, How could the salt destroy the fat?; L405, Bitternes increased as the strain is added earlier… ???). Also, there are statements excessively vague (e.g., L410, bitterness decreased. .., but statistical support is not provided).
L610. Legal limits should also be considered.
Author Response
Response to Reviewer 3 comments
Point 1: Title. The preparation of table olives is diverse. The title should mention the type or style the manuscript is referred to.
Response 1:Title changed according to the suggestion.
Point 2: L20 According?
Response 2: Removed as suggested; the word “Plan” was added.
Point 3: L21. What is regular sweetening? Readers are probably non-specialists in the matter. Explain the meaning.
Response 3: A sentence was added to better explain the traditional process of manufacture of Cobrançosa table olives; the term “regular” was meanwhile removed.
Point 4: L22. It sounds rare that the first season was the last in the explanation. It is certain that the 2019/20 experiment was performed first and, possibly, somehow induced the following ones. Or not?
Response 4: No. In the original plan, trial i106 at the final stage had not been included. However, as experimentation evolved, we realized that plan C was essential to better understand all stages (i.e. sweetening, salting, ready-to-eat). We had to use the table lives from the 2019/20 harvest because of the narrow availability of the taste panel members to meet as a group (June); in fact, olives from the 20/21 harvest from 20/21 would not have been ready before September/October.
Point 5: L30. The sentence is obscure.
Response 5: The sentence was accordingly changed to “… supplementary additive (or adjunct culture)…”.
Point 6: L39-40. The reference is inadequate.
Response 6: The reference was changed as suggested.
Point 7: L42. This reference is also inadequate. Please revise the use of references in the text. They should be related to the issues in context.
Response 7: The reference was changed as suggested.
Point 8: L65-66. The period in water is long. What could be the safety risk level?
Response 8: The sweetening stage is relatively long, as realized from inspection of Figure 2c. This is why we decided to pursue to addition of Lpb. pentosus strain i106 and salting at the very beginning; it is apparent in this Figure that addition of strain i106 makes the numbers of Enterobacteriaceae decrease faster than those in the control. However, this type of table olive is ready to sell (and so to eat) after the salting stage – as the whole process takes 9 to 10 months.
Point 9: L65-67 renewals of brines? When the salt or brine was added?
Response 9: There was a typo from our side; the correct word is water, so we changed it.
Point 10: L65-70. The process should be explained better.
Response 10: To address this request, we added the sentence “The full characterization of processing method and of microbiological and physico-chemical profiles throughout spontaneous fermentation was done by Reis et al. [2022].“ Therefore, if the reader seeks more detailed information about the process, they are directed to the original reference.
Point 11: L129 How were the olives preserved till elaboration?
Response 11: As mentioned in text, spring water was utilized for plans A and B; we further added to the text that delivery took place on the next day upon harvest. For plan C, the word sealed was added regarding the 5% salt brine.
Point 12: L131-147. The experimental design is hard to follow. Furthermore, Fig 1 is also complex, and its contribution to understanding the experiment is limited. One may understand that processing is rather complicated, but a clear explanation is necessary to follow the evolution properly.
Response 12: No comments were received in this regard from the other two reviewers, so no change was implemented; we actually believe that the Figure itself is self-explanatory. To help in grasping said pieces of information, a tutored analysis is provided next.
“According to Figure 1, table olive fermentation from designs A and B were monitored over 35 weeks (245 days), while those from design C were monitored over 10 weeks (65 days) – see the date of harvest as grey arrows, and a significant separation of plans A and B relative to plan C.
In plan A, the addition of strain Lpb. pentosus i106 to 4 independent trials was considered: table olive water with no salt (A0) – obtained right after one week of harvest by the producer, and brines with 3 (A3), 5 (A5), and 7%(w/) (A7) commercial kitchen salt. – see red arrow.
In plan B, the addition of Lpb. pentosus i106 to the brine was tested at 5 (B5) and 7%(w/w) (B7) salt, during the salting process of fermentation – i.e. obtained after sweetening in loco by the producer. – see blue arrow. All information requested is already there.
In plan C, the Lpb. pentosus i106 addition was assessed in ready-to-eat table olives (C5) – i.e. obtained after sweetening and salting in loco by the producer. Although only one trial has been inoculated in each inoculation moment, salt was added to all vessels. – see green arrow.…
For each plan, a control trial was also monitored (AC, BC, and CC) – differing from experimental trials as per addition of the potentially probiotic LAB strain. All trials were run in duplicate.”
All other information has been added to make sure that the design is reproducible.
Point 13: L178. A reference could be pertinent.
Response 13: The sentence already quotes a reference.
“Fatty acids were evaluated by gas chromatography, according to European Commission Regulation (EEC 2568/91, of 11th July), and as described by Rodrigues, et al. [24].”
Point 14: Figure 2-4. Identification of lines should be situated on the top of the plots. Its current situation may induce error. Besides, the meaning of the arrow colours is not explained.
Response 14: Changes were implemented, as suggested. About the arrows, the explanation is at the end of the figures captions.
Point 15: Tables 1-5. The use of average± standard deviation in tables has an uncertain meaning. From the statistical side, it only represents about 68% population. Besides, their values for the same parameters on different samples are diverse. They may cause doubts about such differences' origin (method, matrixes, …). Since the authors applied ANOVA for comparison, it would be more informative to remove the individual values in each average for a pooled standard deviation at the end of each column just above the p-value. Maybe including the mean value for each parameter and the standard deviation (in parenthesis) could also facilitate identifying more/less favourable treatments more straightforwardly. Furthermore, the p-value could be removed since the footnote includes the limit for significance. Eliminating standard values in each row would allow a more straightforward comparison among means.
Response 15: We believe that the tables are understandable by themselves; in fact, we reviewed many papers in attempts to find the best way to represent our results – and they systematically ended up with a representation similar to ours. In fact, no negative comments from the other two reviewers were received on this point. Additionally, all information pertaining to the p-values should be kept because it was retrieved from distinct statistical analyses. The p-value in line came from comparison between trials; while the p-value in column came from the comparison between times. Mention of the p-value in the final part of the figure caption is mandatory: “Means followed by different lowercase letters in each column differed significantly (p < 0.05) among trials for a given sampling time. A-E Means followed by different capital letters in each row differed significantly (p < 0.05) among sampling times for a given trial, as per Tuckey Post-Hoc comparison.”
Finally, we showed fewer sampling points along time in an effort to shorten the original (complete) table. The sampling points are explained in Line 271. More information would likely be redundant with regard to supporting the discussion and conclusions. Anyway, we kept all data and results – so we can add them as a supplementary file, if you so recommend and the editor accepts.
Point 16: Comments to Figures and Table. Please revise the comparisons and hypothesis. Various treatments share the same letters, but many have only one. Then identifying differences could be not straightforward.
Response 16: We revised the text, as suggested.
Point 17: Also, the hypothesis based on parameter values could be misleading (e.g.
L383 fat content tended to decrease with time of addition of probiotic strain and salt concentration; was the fat used by the organisms?, How could the salt destroy the fat?;
Response 17: We agree that the results obtained would hardly support this type of conclusions; therefore, this part was deleted.
Point 18: L405, Bitternes increased as the strain is added earlier… ???).
Response 18: The term “increased” can be misleading; hence, we changed it to “Bitterness of drupes when the strain was added at early processing time of plan A was higher than that when strain was added latter on – 6 for AC and A0, 5 for A3, 4 for A5, and 3 for A7”
This statement can be explained by 1) the profiles of LAB under plan A (figure 2) – as salt increased with strain i106 added later; the strains need more time to re-adapt to the environment and to grow; together, with 2) profiles of oleuropein on BRINE of plan A (figure 5) – as salt increased with strain i106 added later, the concentration of oleuropein is lower in brine (i.e. higher in drupe). All this agrees with the assessment of bitterness as per the taste panel members.
Additionally, it is stated in the text that Lpb. pentosus i106 was checked in term of technological performance (line 108). We re-wrote this sentence as “The aforementioned LAB strain, originally isolated from Cobrançosa table olives [9], was previously tested for its technological properties (such as acid and salt tolerances, survival at different temperatures, and degradation of oleuropein) and probiotic potential in vitro [17,18]”.
Point 19: L410, bitterness decreased. .., but statistical support is not provided).
Response 19: We responded to that query before. About statistical information, we drew confidence intervals in the graphs.
Point 20: L610. Legal limits should also be considered.
Response 20: As far as our knowledge goes, no legal limits have been set forth in Portuguese legislation in this regard.

Round 3
Reviewer 3 Report
Most suggestions were introduced in the revised version. Regarding table presentation, the presentation is common, but this not means to be necessarily correct. What is the statistical meaning of mean±standard deviation? Authors should think about it.